# IMPACT: Influence Modeling for Open-Set Time Series Anomaly Detection

Xiaohui Zhou [1 2]   Yijie Wang [1 2 *]   Hongzuo Xu [3]   Weixuan Liang [2]   Xiaoli Li [4]   Guansong Pang [5 *]

## Abstract

Open-set anomaly detection (OSAD) is an emerging paradigm designed to utilize limited labeled data from anomaly classes seen in training to identify both seen and unseen anomalies during testing. Current approaches rely on simple augmentation methods to generate pseudo anomalies that replicate unseen anomalies. Despite being promising in image data, these methods are found to be ineffective in time series data due to the failure to preserve its sequential nature, resulting in trivial or unrealistic anomaly patterns. They are further plagued when the training data is contaminated with unlabeled anomalies. This work introduces **IMPACT**, a novel framework that leverages **i**nfluence **m**odeling for o**p**en-set time series **a**nomaly dete**ct**ion, to tackle these challenges. The key insight is to **i)** learn an influence function that can accurately estimate the impact of individual training samples on the modeling, and then **ii)** leverage these influence scores to generate semantically divergent yet realistic unseen anomalies for time series while repurposing high-influential samples as supervised anomalies for anomaly decontamination. Extensive experiments show that IMPACT significantly outperforms existing state-of-the-art methods, showing superior accuracy under varying OSAD settings and contamination rates. Code is available at https://github.com/mala-lab/IMPACT.

---

[1]National Key Laboratory of Parallel and Distributed Computing [2]College of Computer Science and Technology, National University of Defense Technology [3]Intelligent Game and Decision Lab (IGDL) [4]Information Systems Technology and Design, Singapore University of Technology and Design [5]School of Computing and Information Systems, Singapore Management University. Correspondence to: Yijie Wang <wangyijie@nudt.edu.cn>, Guansong Pang <gspang@smu.edu.sg>.

*Proceedings of the 43rd International Conference on Machine Learning*, Seoul, South Korea. PMLR 306, 2026. Copyright 2026 by the author(s).

## 1. Introduction

Time series anomaly detection (TSAD) aims at identifying unusual patterns that significantly deviate from the majority of the data characterized by sequential observations over time (Blázquez-García et al., 2021), and has been widely applied in critical domains ranging from financial fraud detection (Anandakrishnan et al., 2018) to healthcare (Pereira & Silveira, 2019) and industrial equipment monitoring (Wang et al., 2022). Since it is impractical to obtain extensive labeled anomalies in real-world scenarios, most existing TSAD approaches (Liu & Paparrizos, 2024; Wu et al., 2025; Shentu et al., 2025) are unsupervised, assuming that only normal data is available during training. However, some labeled anomaly samples are often accessible in many applications, such as abnormal transactions identified in the past financial management and electrocardiograms (ECG) of patients. These anomaly samples offer important priors about domain-specific abnormality, but the unsupervised methods are unable to utilize them.

Recently, open-set anomaly detection (OSAD) (Ding et al., 2022; Acsintoae et al., 2022; Zhu et al., 2024; Wang et al., 2025) is proposed to leverage the limited labeled anomaly samples to learn generalized models for detecting both seen anomalies from closed-set anomaly classes and unseen anomalies from open-set anomaly classes. While OSAD can utilize prior knowledge from the labeled data to reduce false positive errors, this paradigm still faces two key challenges when tailored for time series data. First, aside from a limited set of labeled anomalies, current methods (Pang et al., 2019; 2023; Lai et al., 2024; Liu et al., 2025) conventionally treat the remaining unlabeled training samples as normal to facilitate supervised training. However, the presence of unknown anomalies in the training set (*i.e.*, anomaly contamination) is unavoidable (Pang et al., 2023; Xu et al., 2024), which can largely mislead the training.

Second, pseudo anomalies are typically generated through data augmentation (Lappas et al., 2024; Dong et al., 2024) to mimic unseen anomalies to enhance its generalization. Nevertheless, the mainstream augmentation operations such as rotation and cropping are image data oriented, rendering them ineffective in generating realistic pseudo time-series anomalies. Low-quality pseudo anomalies may introduce misleading signals that are either indistinguishable from

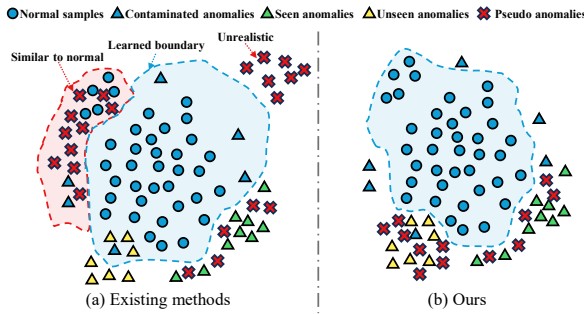

*Figure 1.* **(a)** Contaminated anomalies and low-quality pseudo anomalies significantly affect the learned boundary. **(b)** IMPACT addresses these issues by i) accurately flipping the contaminated unlabeled anomalies into labeled ones through influence modeling and ii) the influence-score-guided generation of high-quality pseudo anomalies from the perspective of reducing the test risk.

normal patterns or entirely unrepresentative of real anomalies. For instance, short-range moving averages (Goswami et al., 2023) may fail to distort long-term seasonal patterns effectively, while horizontally flipping a segment (Obata et al., 2025) in ECG data would produce a pattern that violates fundamental cardiac electrophysiology. Consequently, existing methods can be misled to biased decision boundary, leading to high false positives and missed detection of unseen anomalies, as shown in Figure 1(a).

To address these challenges, we propose **IMPACT**, a novel framework that leverages **i**nfluence **m**odeling for o**p**en-set time series **a**nomaly dete**ct**ion. It leverages the theoretical properties of influence functions (Hampel, 1974) to systematically purify the training data and synthesize high-quality pseudo anomalies, as illustrated in Figure 1(b). To this end, IMPACT is instantiated with two novel modules, including *Test-risk-driven Influence Scoring (TIS)* and *Risk-reduction-based Anomaly Decontamination and Generation (RADG)*. TIS first performs a multi-channel deviation loss-based influence modeling to precisely quantify the effect of each training sample on the model's test risk. We theoretically show that minimizing this loss is equivalent to the entropy minimization of the latent distribution, encouraging the model to learn a compact representation for normal data while maximizing the deviation of anomalies. The resulting influence scores are then utilized by the RADG module from the perspective of reducing the test risk to i) repurpose contaminated samples exhibiting high influence scores as true labeled anomalies and ii) select and perturb boundary normal samples that contribute least to risk minimization to generate pseudo anomalies that effectively simulate unseen anomaly patterns under theoretical guarantee.

Our main contributions are summarized as follows:

- We propose IMPACT, the first framework to leverage influence modeling for open-set TSAD, simultaneously

addressing the dual challenges of anomaly contamination and generation in time series data.

- We further instantiate IMPACT with theoretically grounded TIS and RADG modules, which accurately quantify and utilize influence scores to repurpose contaminated samples as labeled anomalies while generating semantically realistic unseen anomalies.

- Extensive experiments on multiple real-world datasets demonstrate that IMPACT achieves state-of-the-art performance, exhibiting superior robustness against contamination and effective detection of unseen anomalies compared to existing baselines.

## 2. Related Work

**Time Series Anomaly Detection.** Existing TSAD methods (Shentu et al., 2025; Ho et al., 2025) are predominantly designed as unsupervised learning due to the difficulty of collecting massive anomalies. The most popular approaches (Zhou et al., 2023; Wu et al., 2025) learn to reconstruct the entire series through an encoder-decoder framework, and anomalies are difficult to reconstruct accurately. One-class classification methods (Shen et al., 2020; Xu et al., 2024) focus on learning data normality using support vectors. Other representative approaches include prediction and self-supervised learning methods (Yang et al., 2023; Ansari et al., 2024; Das et al., 2024). Nevertheless, these methods are unable to leverage any prior knowledge about real anomalies, leading to high false positives.

**Open-Set Anomaly Detection.** OSAD methods (Ding et al., 2022; Xu et al., 2023; Lai et al., 2024; Shao et al., 2025) seek to reduce the detection errors by utilizing limited labeled anomalies from partial anomaly classes that are often available in real-world applications. DRA (Ding et al., 2022) learns disentangled representations of seen, pseudo, and residual anomalies to enhance the detection of both seen and unseen anomalies. AHL (Zhu et al., 2024) utilizes seen and pseudo anomalies to simulate heterogeneous anomaly distributions for generalization. DPDL (Wang et al., 2025) encloses normal samples within compact distribution space while steering anomalies away. These methods rely on augmentation methods to generate pseudo anomalies that mimic unseen anomalies. Although they are effective in image data, recent approaches such as MOSAD (Lai et al., 2024) and InvAD (Liu et al., 2025) still struggle to generate pseudo anomalies tailored for time series data. Recently, WSAD-DT (Durani et al., 2025) proposes a dual-tailed kernel mechanism to handle the diversity of abnormal behaviors.

**Data Augmentation for Anomaly Generation.** Data augmentation can generate pseudo anomalies to improve the generalization to unseen anomalies. Many augmentation methods are proposed in computer vision, such as Mixup

(Zhang et al., 2018), CutMix (Yun et al., 2019), and CutPaste (Li et al., 2021a). DDL (Lappas et al., 2024) introduces dynamic anomaly weighting to refine pseudo anomaly patterns. However, these image-tailored augmentations are difficult to be directly applied to time series. Inspired by image operations, COE and WMix are proposed to generate time series-specific augmentations via segment swapping and combination (Carmona et al., 2022). COUTA (Xu et al., 2024) utilizes data perturbation to create native anomaly samples. CutAddPaste (Wang et al., 2024) synthesizes diverse anomalies by mimicking five distinct anomaly categories. DADA (Shentu et al., 2025) employs eight data augmentations like flip and scale to inject anomaly in pre-training. However, these methods predominantly rely on heuristic rules and pre-defined priors to synthesize anomalies, which inherently limit the generated pseudo anomalies to variations of seen patterns, rendering them insufficient to simulate the complex distributions of truly unseen anomalies that lie outside these heuristic constraints.

**Contamination Estimation and Anomaly Score Reliability.** Real-world unlabeled training data frequently contains a small fraction of anomalies, known as anomaly contamination (Xu et al., 2024; Zhang et al., 2026). GammaGMM (Perini et al., 2023) infers the contamination rate from unlabeled data for principled threshold selection, while ExCeeD (Perini et al., 2020) and calibration-conditional conformal p-values (Bates et al., 2023) address score reliability through probability estimation and distribution-free hypothesis testing, respectively. However, they primarily operate post-hoc, either estimating contamination after training or calibrating scores at inference time, and do not directly correct contaminated training samples. IMPACT addresses contamination proactively at the training level through influence-guided label flipping, eliminating the detrimental effect of contaminated samples before model optimization.

**Influence-based Data Analysis.** Influence analysis (Koh & Liang, 2017; Hammoudeh & Lowd, 2024) quantifies how individual training samples affect model behavior, and has been applied to data valuation (Ghorbani & Zou, 2019) and data cleaning (Hammoudeh & Lowd, 2024). However, existing approaches treat these as separate tasks and do not leverage influence information to synthesize new samples. IMPACT goes beyond this by introducing a dual-purpose influence scoring mechanism that simultaneously performs decontamination via label flipping and pseudo anomaly generation via influence-guided feature perturbation, both within a unified risk-reduction framework.

## 3. Proposed Approach

**Problem Statement.** Let $\mathcal{D} = \{(\boldsymbol{x}_i, y_i) \in \mathcal{X} \times \mathcal{Y}\}_{i=1}^N$ be the time series training set for the studied open-set problem. In contrast to unsupervised TSAD where no labeled anoma-

lies from the target dataset are utilized, the open-set setting assumes the availability of a minimal number of labeled anomaly samples. Here, $\mathcal{D}_n$ ($|\mathcal{D}_n| = M$) is the normal subset with potential anomaly contamination (*i.e.*, a minimal fraction of samples deemed normal ($y_i = 0$) may actually be abnormal ($y_i = 1$)) and $\mathcal{D}_a$ ($|\mathcal{D}_a| = A$) is a very small set of labeled anomalies from *seen* anomaly classes. Formally $\mathcal{D} \doteq \mathcal{D}_n \cup \mathcal{D}_a$, where $A \ll M$. $\boldsymbol{x}_i \in \mathbb{R}^{D \times L}$ denotes the $i^{th}$ time interval, which is collected from $D$ variables over $L$ consecutive timestamps. Each time interval is regarded as a sample. The labeled anomalies in $\mathcal{D}_a$ belong to a set of seen anomaly classes $\mathcal{S} \subseteq \mathcal{C}$, where $\mathcal{C} = \{c_i\}_{i=1}^{|\mathcal{C}|}$ denotes the set of all possible anomaly classes. The remaining classes $\mathcal{U} = \mathcal{S} \setminus \mathcal{C}$ are *unseen* anomaly classes, for which no labeled samples are available during training. Note that, the contaminated data in $\mathcal{D}_n$, *i.e.*, *unlabeled anomalies*, may be from either $\mathcal{S}$ or $\mathcal{U}$ that are mislabeled as normal, since in real-world scenarios the normal subset is collected without any prior knowledge of whether or to what degree it has been contaminated by anomalies of any class. Our goal is to learn an anomaly score function $f : \mathcal{X} \to \mathbb{R}$ that assigns larger anomaly scores to anomalies spanning all anomaly classes than normal samples.

**Overview.** Figure 2 provides an overview of IMPACT, which consists of three major phases:

**Phase I**: **Test-risk-driven Influence Scoring** (**TIS**) module first trains an initial model by minimizing the empirical risk over all training samples $\boldsymbol{z}_i = (\boldsymbol{x}_i, y_i)$, and get the optimized parameters defined as: $\hat{\theta} = \arg\min_{\theta \in \Theta} \frac{1}{N} \sum_{i=1}^N L(f(\boldsymbol{x}_i, \theta), y_i)$, where $L$ is the multi-channel deviation loss described in Section 3.1. We denote $L(f(\boldsymbol{x}_i, \theta), y_i) = L(\boldsymbol{z}_i, \theta)$ for short. Initial model $f(\cdot, \theta)$ can be defined as a combination of a feature extractor $\phi(\cdot, \theta_\phi)$ and a learning head $h(\cdot, \theta_h)$, formalized as: $f(\cdot, \theta) = h(\phi(\cdot, \theta_\phi), \theta_h)$, where $\theta = \{\theta_\phi, \theta_h\}$. Then TIS models the influence of perturbing each training sample on the test risk, and utilizes these resulting influence scores to select contaminated samples $\mathcal{D}_{con}$, clean samples $\mathcal{D}_{clean}$ and normal reference samples $\mathcal{D}_{ref}$ from $\mathcal{D}$.

**Phase II**: **Risk-reduction-based Anomaly Decontamination and Generation** (**RADG**) module leverages the influence scores from a risk-reduction perspective to simultaneously eliminate contamination and expand the anomaly manifold. For **anomaly decontamination**, RADG performs label flipping on each identified contaminated sample in $\mathcal{D}_{con}$, and obtains an additional set of true anomalies denoted as $\mathcal{D}'_{con}$. We theoretically prove that updating the model parameters by introducing $\mathcal{D}'_{con}$ can achieve lower test risk. For **anomaly generation**, RADG selects the $k$ normal samples from $\mathcal{D}_{clean}$ that contribute the least to risk minimization. The underlying intuition is that these samples' features lack sufficient normality to guide the model

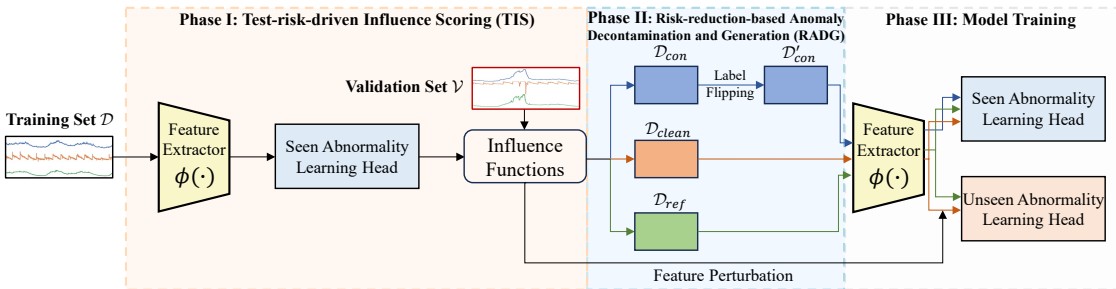

*Figure 2.* The overall framework of IMPACT. TIS first quantifies the influence of training samples on test risk, then RADG leverages these influence scores from the risk-reduction perspective to decontaminate the training set via label flipping and synthesize realistic unseen anomalies through feature perturbation. A dual-head architecture is finally trained to effectively detect both seen and unseen anomalies.

training, residing near the normality-abnormality boundary. This makes them more susceptible to being affected by perturbations to generate new abnormal characteristics. We theoretically prove that the new features obtained through perturbation introduce unseen abnormality that does not follow the known patterns, and further reduce the test risk.

**Phase III**: In addition to learning seen abnormality from limited given anomalies, the final training adds an unseen abnormality learning head to generalize unseen anomalies.

### 3.1. Test-risk-driven Influence Scoring

**Initial Training.** We conduct initial training based on training samples $z_i = (x_i, y_i)$. Inspired by the deviation loss (Pang et al., 2019), we first formalize multi-channel deviation scores as follows:

$$dev(x_i) = \sqrt{(f(x_i, \theta) - \mu_r)^\top \Sigma_r^{-1}(f(x_i, \theta) - \mu_r)}, \quad (1)$$

where $f(x_i, \theta)$ outputs multi-channel anomaly scores $s_i \in \mathbb{R}^r$ indicating the likelihood of a sample being abnormal, $\mu_r$ and $\Sigma_r$ are the mean and covariance of the joint Gaussian prior-based anomaly score set $\{r_1, r_2, \cdots, r_l\}$. Each $r_i$ is drawn from $\mathcal{N}(\mu, \Sigma)$. We choose isotropic Gaussian to achieve stable detection performance on various datasets. Then we propose a multi-channel deviation loss to optimize model, which can be formalized as follows:

$$L(z_i, \theta) = \frac{1}{r}\sum_{j=1}^{r}[(1-y_i)dev(x_i)_j + y_i \max\left(0, a - dev(x_i)_j\right)]. \quad (2)$$

Geometrically, this loss encourages the model to push the anomaly scores of normal samples towards $\mu_r$, while enforcing the anomaly scores of anomalies must deviate from $\mu_r$ by at least $a$. The key idea behind this is to evaluate the abnormality of samples from multiple dimensions, which is analogous to assessing an object from multiple angles. For normal objects, every perspective should align with expectations, but for abnormal objects, there will always be one perspective that stands out as abnormal. Next, we explain that this geometric interpretation can also be understood as

entropy minimization in the latent distribution. We derive the Theorem 1, and the proof is provided in Appendix A.

**Theorem 1.** *Assuming the multi-channel anomaly scores $s \in \mathbb{R}^r$ follow a latent distribution $S \sim \mathcal{N}(\mu, \sigma^2 I)$, the geometric proximity to the isotropic Gaussian prior is equivalent to the entropy minimization of the latent distribution:*

$$\mathcal{H}(S) = \frac{r}{2}(1 + \log(2\pi\sigma^2)) \propto \log \sigma^2. \quad (3)$$

Since the entropy of $S$ is proportional to its log-variance, Theorem 1 indicates that the entropy minimization encourages the model to make its predictions (the anomaly scores) as close as possible to the isotropic Gaussian prior that represents the normal data distribution.

**Influence Modeling.** Intuitively, if discarding a sample from training set would reduce the test risk, it indicates that this sample is detrimental to the model optimization, and is most likely a contaminated sample. Based on influence functions (Hampel, 1974; Koh & Liang, 2017), we can measure how each training sample affects test risk.

When perturbing $z_i$ by small $\epsilon_i$, we obtain new parameters $\hat{\theta}_{\epsilon_i} = \arg\min_{\theta \in \Theta} \frac{1}{N}\sum_{n=1}^{N} L(z_n, \theta) + \epsilon_i L(z_i, \theta)$, and the change in model parameters can be estimated as:

$$\mathcal{I}_\theta(z_i) \triangleq \frac{d\hat{\theta}_{\epsilon_i}}{d\epsilon_i}|_{\epsilon_i=0} = -H_{\hat{\theta}}^{-1}\nabla_\theta L(z_i, \hat{\theta}), \quad (4)$$

where $H_{\hat{\theta}} \triangleq \frac{1}{N}\sum_{n=1}^{N} \nabla_\theta^2 L(z_n, \hat{\theta})$ is the Hessian matrix. Using the chain rule, we have the following closed-form expression to estimate the influence of perturbing $z_i$ on the prediction of a test sample $z_t$:

$$\mathcal{I}_L(z_i, z_t) \triangleq \frac{dL(z_t, \hat{\theta}_{\epsilon_i})}{d\epsilon_i}|_{\epsilon_i=0} \quad (5)$$
$$= -\nabla_\theta L(z_t, \hat{\theta})^\top H_{\hat{\theta}}^{-1} \nabla_\theta L(z_i, \hat{\theta}).$$

Therefore, if we perturb $z_i$ by small $\epsilon_i$, we can estimate the change of loss at a test sample $z_t$ taken from the validation set $\mathcal{V}$ as follows:

$$L(z_t, \hat{\theta}_{\epsilon_i}) - L(z_t, \hat{\theta}) \approx \epsilon_i \times \mathcal{I}_L(z_i, z_t). \quad (6)$$

Furthermore, we can estimate the influence of perturbing $z_i$ on the test risk over $\mathcal{V}$ as follows:

$$L(\mathcal{V}, \hat{\theta}_{\epsilon_i}) - L(\mathcal{V}, \hat{\theta}) \approx \epsilon_i \times \sum_{z_t \in \mathcal{V}} \mathcal{I}_L(z_i, z_t), \quad (7)$$

where a negative result indicates that the test risk after the perturbation is lower than before the perturbation. Inspired by (Kong et al., 2022), noting that given $\epsilon_i \in [-\frac{1}{N}, 0)$, Equation (7) actually estimates the influence of downweighting or discarding the training sample $z_i$. Therefore, when influence score $\mathcal{I}_L(z_i) = \sum_{z_t \in \mathcal{V}} \mathcal{I}_L(z_i, z_t) > 0$, it indicates that $z_i$ is harmful to model's prediction. We can select contaminated samples $\mathcal{D}_{con} = \{z_i \in \mathcal{D}_n \mid \mathcal{I}_L(z_i) > 0\}$. On the contrary, when $\mathcal{I}_L(z_i)$ is less than 0, it indicates that the sample is beneficial for the model's prediction. We can select the $k$ most helpful samples as normal reference samples $\mathcal{D}_{ref} = \{z_i \in \mathcal{D}_n \mid \mathcal{I}_L(z_i) < 0\}|_{\text{top-k smallest}}$, which will be used to further amplify the deviation between normality and abnormality. The remaining normal and abnormal labeled samples form a clean set $\mathcal{D}_{clean} = \mathcal{D} \setminus (\mathcal{D}_{con} \cup \mathcal{D}_{ref})$.

### 3.2. Risk-reduction-based Anomaly Decontamination and Generation

**Anomaly Decontamination via Label Flipping.** We propose to flip the label of each contaminated sample in $\mathcal{D}_{con}$. First, we consider the influence of perturbing label from $z_i = (x_i, y_i)$ to $z_{i\delta_i} = (x_i, y_i + \delta_i)$. According to (Koh & Liang, 2017), the new parameters after moving $\epsilon_i$ mass from $z_i$ to $z_{i\delta_i}$ can be defined as: $\hat{\theta}_{\epsilon_i \delta_i} = \arg\min_{\theta \in \Theta} \frac{1}{N} \sum_{n=1}^{N} L(z_n, \theta) + \epsilon_i L(z_{i\delta_i}, \theta) - \epsilon_i L(z_i, \theta)$. The closed-form estimate of the effect of $z_i \mapsto z_{i\delta_i}$ on model parameters can be formalized as follows:

$$\frac{d\hat{\theta}_{\epsilon_i \delta_i}}{d\epsilon_i}\Big|_{\epsilon_i=0} = \mathcal{I}_\theta(z_{i\delta_i}) - \mathcal{I}_\theta(z_i)$$
$$= -H_{\hat{\theta}}^{-1}(\nabla_\theta L(z_{i\delta_i}, \hat{\theta}) - \nabla_\theta L(z_i, \hat{\theta})). \quad (8)$$

We can further estimate the influence of perturbing $z_i$ by $z_{i\delta_i}$ on the prediction of a test sample $z_t$:

$$\mathcal{I}_{L\delta_i}(z_i, z_t) = \frac{dL(z_t, \hat{\theta}_{\epsilon_i \delta_i})}{d\epsilon_i}\Big|_{\epsilon_i=0}$$
$$= -\nabla_\theta L(z_t, \hat{\theta})^\top H_{\hat{\theta}}^{-1}(\nabla_\theta L(z_{i\delta_i}, \hat{\theta}) - \nabla_\theta L(z_i, \hat{\theta})). \quad (9)$$

Equation (9) offers a way to measure the influence of changing the labels of training samples on the model's predictions.

Then we analyse the test risk would be reduced by converting $z_i = (x_i, 0) \in \mathcal{D}_{con}$ to $z_{i\mathbf{1}} = (x_i, 1) \in \mathcal{D}'_{con}$. Without loss of generality, we set $\mu_r = 0$ and $\Sigma_r = I$, and treat the output of $f(x_i, \theta)$ as a single-channel scalar (the multi-channel result is the average). The loss is simplified as: $L(z_i, \theta) = (1 - y_i) f(x_i, \theta) + y_i \max(0, a - f(x_i, \theta))$.

When model is retrained with $\mathcal{D}'_{con}$, We disregard the case where $a - f(x_i, \theta) < 0$, as it does not result in parameter updates, and the loss $L(z_i, \theta)$ is changed from $f(x_i, \theta)$ to $a - f(x_i, \theta)$. We have:

$$\nabla_\theta L(z_{i\mathbf{1}}, \theta) - \nabla_\theta L(z_i, \theta) = -2\nabla_\theta L(z_i, \theta). \quad (10)$$

Substituting Equation (10) into Equation (9), and combing it with Equation (5), the influence of flipping label of $z_i$ on the prediction at test sample $z_t$ is:

$$\mathcal{I}_{L\mathbf{1}}(z_i, z_t) = 2\nabla_\theta L(z_t, \hat{\theta})^\top H_{\hat{\theta}}^{-1} \nabla_\theta L(z_i, \hat{\theta})$$
$$= -2\mathcal{I}_L(z_i, z_t). \quad (11)$$

Similar to Equation (7), we denote the optimal parameters as $\hat{\theta}_{\epsilon_i \mathbf{1}}$ after flipping label of $z_i$, and extend the influence to the whole test risk over $\mathcal{V}$:

$$L(\mathcal{V}, \hat{\theta}_{\epsilon_i \mathbf{1}}) - L(\mathcal{V}, \hat{\theta}) \approx \epsilon_i \times \sum_{z_t \in \mathcal{V}} \mathcal{I}_{L\mathbf{1}}(z_i, z_t). \quad (12)$$

According to Equation (11), the influence of label flipping is related to the influence of perturbing $z_i$. Then we derive the Theorem 2, and the proof is provided in Appendix B.

**Theorem 2.** *Label flipping on the contaminated samples in $\mathcal{D}_{con} = \{z_i \in \mathcal{D}_n \mid \mathcal{I}_L(z_i) > 0\}$ can achieve lower test risk over $\mathcal{V}$:*

$$L(\mathcal{V}, \hat{\theta}_{\epsilon\mathbf{1}}) - L(\mathcal{V}, \hat{\theta}) \approx -\frac{2}{N \cdot |\mathcal{D}_{con}|} \sum_{z_i \in \mathcal{D}_{con}} \mathcal{I}_L(z_i) < 0. \quad (13)$$

Theorem 2 shows that flipping the label of each contaminated sample in $\mathcal{D}_{con}$ can reduce the test risk. The benefit of label flipping lies in its ability to not only avoid the negative effects of anomaly contamination but also transform it into valuable abnormal knowledge that can be leveraged, ultimately improving the model's performance.

**Anomaly Generation via Feature Perturbation.** Then we focus on generalizing the model's ability to detect previously unseen anomalies by perturbing the features of clean samples. To implement this, we first utilize influence scores to select the $k$ normal samples $\mathcal{D}_{per}$ in the clean set $\mathcal{D}_{clean}$ that contribute the least to risk minimization, defined as $\mathcal{D}_{per} = \{z_i \in (\mathcal{D}_n \cap \mathcal{D}_{clean}) \mid \mathcal{I}_L(z_i) < 0\}|_{\text{top-k largest}}$. The underlying hypothesis is that these least helpful samples are more prone to being influenced by perturbations, thus offering an opportunity to generate new types of abnormality when modified.

Unlike traditional methods (Carmona et al., 2022; Lai et al., 2024; Wang et al., 2024) that manually design transformations to generate pseudo anomalies, which may not always guarantee the quality of the generated anomalies, we instead generate unseen anomalies directly from the feature space based on the influence functions. Recall that

our model $f(\cdot, \theta)$ includes feature extractor $\phi(\cdot, \theta_\phi)$ and learning head $h(\cdot, \theta_h)$, and let $\varphi_i = \phi(x_i, \theta_\phi)$. Then we can obtain the labeled feature set $\mathcal{W}_{per} = \{w_i = (\varphi_i, 0) \mid (x_i, 0) \in \mathcal{D}_{per}\}$. Consider the feature perturbation $w_{i\zeta_i} = (\varphi_i + \zeta_i, 0)$, and we can define the optimal learning head parameters after moving $\epsilon_i$ mass from $w_i$ to $w_{i\zeta_i}$: $\hat{\theta}_{h,\epsilon_i\zeta_i} = \arg\min_{\theta_h \in \Theta_h} \frac{1}{N} \sum_{n=1}^{N} L(w_n, \theta_h) + \epsilon_i L(w_{i\zeta_i}, \theta_h) - \epsilon_i L(w_i, \theta_h)$. Similar to Equation (9), we can further approximate the effect of $w_i \mapsto w_{i\zeta_i}$ on the loss at test feature $w_t$:

$$
\begin{aligned}
\mathcal{I}_{L\zeta_i}(w_i, w_t) &= \frac{dL(w_t, \hat{\theta}_{h,\epsilon_i\zeta_i})}{d\epsilon_i}\Big|_{\epsilon_i=0} \\
&\approx -\nabla_{\theta_h} L(w_t, \hat{\theta}_h)^\top H_{\hat{\theta}_h}^{-1} [\nabla_\varphi \nabla_{\theta_h} L(w_i, \hat{\theta}_h)]\zeta_i.
\end{aligned}
\tag{14}
$$

Then the influence of $w_i \mapsto w_{i\zeta_i}$ on the test risk over $\mathcal{V}$ can be estimated as:

$$
\begin{aligned}
\mathcal{I}_{L\zeta_i}(w_i) &= \sum_{w_t \in \mathcal{V}} \mathcal{I}_{L\zeta_i}(w_i, w_t) \\
&\approx \underbrace{-\nabla_{\theta_h} L(\mathcal{V}, \hat{\theta}_h)^\top H_{\hat{\theta}_h}^{-1} [\nabla_\varphi \nabla_{\theta_h} L(w_i, \hat{\theta}_h)]}_{\mathcal{I}_{per}(w_i)} \zeta_i.
\end{aligned}
\tag{15}
$$

According to Equation (15), we can set $\zeta_i$ in the direction of $\mathcal{I}_{per}(w_i)^\top$ to construct feature perturbations of $w_i$ that maximally increase test risk. The intuition is that when the feature perturbation causes the greatest increase in test risk, it indicates that the perturbed features have significantly deviated from the original label attributes, *i.e.*, they are in an unknown abnormal distribution. Then we can construct perturbed feature set $\mathcal{W}'_{per} = \{w_{i\zeta_i\mathbf{1}} = (\varphi_{i\zeta_i}, 1) \mid (x_i, 0) \in \mathcal{D}_{per}\}$, where $\varphi_{i\zeta_i} = \varphi_i + \alpha \mathcal{I}_{per}(w_i)^\top$, and $\alpha > 0$ is the perturbation strength. Now we derive the Theorem 3, and the proof can be found in Appendix C.

**Theorem 3.** *Considering the original feature $\varphi_i$ and the feature perturbation $\alpha \mathcal{I}_{per}(w_i)^\top$. Measuring in the feature space, the perturbed feature $\varphi_{i\zeta_i}$ follows a new distribution that is different from the original distribution of $\varphi_i$, and there exists a lower bound for the distribution difference.*

Theorem 3 shows that the distribution of perturbed feature can be very different from that of the original feature under the influence-guided perturbation, which further promotes the model's generalization ability to unseen anomalies. Similar to Equation (12), we derive the Theorem 4 to show that the perturbed features can further reduce the test risk, and the proof is provided in Appendix D.

**Theorem 4.** *Generating anomalies via feature perturbation $\alpha \mathcal{I}_{per}(w_i)^\top$ from feature set $\mathcal{W}_{per}$ can further reduce the test risk over $\mathcal{V}$:*

$$
\begin{aligned}
&L(\mathcal{V}, \hat{\theta}_{h,\epsilon_i\zeta_i\mathbf{1}}) - L(\mathcal{V}, \hat{\theta}_h) \\
&\approx -\frac{\alpha}{N \cdot |\mathcal{W}_{per}|} \sum_{w_i \in \mathcal{W}_{per}} \|\mathcal{I}_{per}(w_i)\|_2^2 < 0.
\end{aligned}
\tag{16}
$$

### 3.3. Training and Inference

We perform final training using the augmented datasets comprising the label-flipped contaminated samples $\mathcal{D}'_{con}$, the normal reference samples $\mathcal{D}_{ref}$, the remaining clean samples $\mathcal{D}_{clean}$, and the perturbed feature set $\mathcal{W}'_{per}$ representing unseen anomalies. In addition to the existing parameters $\theta = \{\theta_\phi, \theta_h\}$, we introduce a dedicated unseen abnormality learning head $h'(\cdot, \theta_{h'})$. The final training objective consists of two complementary loss functions.

**Seen Abnormality Learning** focuses on learning from normal samples and identified anomalies (including originally labeled ones and discovered contaminated samples). We aggregate these samples into $\mathcal{D}_s = \mathcal{D}'_{con} \cup \mathcal{D}_{ref} \cup \mathcal{D}_{clean}$:

$$
L_{seen} = \sum_{z_i \in \mathcal{D}_s} L(z_i, \theta).
\tag{17}
$$

**Unseen Abnormality Learning** aims to generalize the model's detection capability to previously unseen anomaly classes. We utilize the most helpful normal samples $\mathcal{D}_h = \mathcal{D}_{ref} \cup (\mathcal{D}_n \cap \mathcal{D}_{clean}) \setminus \mathcal{D}_{per}$ and the perturbed features $\mathcal{W}'_{per}$ that embody unseen abnormal characteristics:

$$
L_{unseen} = \sum_{z_i \in \mathcal{D}_h} L(z_i, \theta) + \sum_{w_{i\zeta_i\mathbf{1}} \in \mathcal{W}'_{per}} L(w_{i\zeta_i\mathbf{1}}, \theta_{h'}).
\tag{18}
$$

The complete training objective combines both components to jointly optimize the model parameters:

$$
L_{re} = L_{seen} + \lambda L_{unseen},
\tag{19}
$$

where $\lambda$ denotes a weight adapting coefficient.

During inference, we compute the final anomaly score for a test sample $x_i$ by combining two scoring functions.

**Maximum Abnormality Score.** We first combine predictions from the seen and unseen abnormality learning heads:

$$
s_m = \max_{l<r} \left(h(\phi(x_i, \theta_\phi), \theta_h) + h'(\phi(x_i, \theta_\phi), \theta_{h'})\right)_l.
\tag{20}
$$

The maximum operation across the $r$ channels ensures that if any channel indicates strong abnormality, the sample will receive a high anomaly score, consistent with our multi-perspective anomaly assessment philosophy.

**Feature Deviation Score.** To detect subtle anomalies, we calculate the deviation of the test sample's features from the normal feature distribution, as represented by the reference normal samples:

$$
s_f = \|\phi(x_i, \theta_\phi) - \frac{1}{|\mathcal{D}_{ref}|} \sum_{x_j \in \mathcal{D}_{ref}} \phi(x_j, \theta_\phi)\|^2.
\tag{21}
$$

The final anomaly score is obtained by combining both components: $s = s_m + s_f$.

*Table 1.* Detection accuracy (AUC $\pm$ standard deviation) of IMPACT and its competing methods under the unsupervised and open-set general settings. All results are in %. The best ones are **boldfaced** and the second best are underlined.

| | Methods | UCR | ASD | PSM | SMD | CT | SAD | PTBXL | TUSZ |
|---|---|---|---|---|---|---|---|---|---|
| Unsupervised | TCN-AE | $50.63_{\pm0.46}$ | $58.73_{\pm0.49}$ | $58.49_{\pm0.79}$ | $64.76_{\pm0.19}$ | $50.02_{\pm0.05}$ | $50.01_{\pm0.02}$ | $58.74_{\pm0.28}$ | $64.17_{\pm0.06}$ |
| | THOC | $52.75_{\pm0.56}$ | $55.74_{\pm1.57}$ | $58.02_{\pm1.51}$ | $62.55_{\pm0.47}$ | $74.50_{\pm3.16}$ | $57.76_{\pm3.14}$ | $62.89_{\pm1.97}$ | $62.45_{\pm1.92}$ |
| | TranAD | $49.26_{\pm0.73}$ | $56.86_{\pm0.68}$ | $60.34_{\pm0.40}$ | $64.40_{\pm0.25}$ | $29.28_{\pm3.12}$ | $49.10_{\pm1.09}$ | $58.51_{\pm1.18}$ | $62.82_{\pm0.73}$ |
| | DCdetector | $51.40_{\pm1.43}$ | $47.94_{\pm1.05}$ | $50.51_{\pm0.72}$ | $50.23_{\pm0.85}$ | $49.20_{\pm1.80}$ | $49.02_{\pm1.15}$ | $51.28_{\pm0.78}$ | $50.55_{\pm1.10}$ |
| | GPT4TS | $54.60_{\pm0.87}$ | $62.42_{\pm2.06}$ | $61.14_{\pm0.63}$ | $66.20_{\pm0.31}$ | $60.32_{\pm0.80}$ | $48.09_{\pm3.59}$ | $47.82_{\pm0.78}$ | $66.31_{\pm0.31}$ |
| | COUTA | $53.44_{\pm1.14}$ | $61.74_{\pm2.67}$ | $58.29_{\pm3.90}$ | $60.93_{\pm1.36}$ | $51.09_{\pm0.00}$ | $50.00_{\pm0.00}$ | $56.71_{\pm1.86}$ | $62.45_{\pm0.89}$ |
| | DADA | $50.06_{\pm0.42}$ | $50.67_{\pm0.09}$ | $54.27_{\pm0.06}$ | $64.45_{\pm0.07}$ | $56.08_{\pm0.12}$ | $49.98_{\pm0.90}$ | $63.22_{\pm0.21}$ | $44.20_{\pm0.01}$ |
| Open-Set | DevNet | $49.82_{\pm0.36}$ | $49.41_{\pm3.41}$ | $49.88_{\pm2.20}$ | $49.62_{\pm1.37}$ | $49.63_{\pm0.65}$ | $50.00_{\pm0.01}$ | $50.00_{\pm0.00}$ | $49.63_{\pm0.80}$ |
| | DSAD | $51.99_{\pm1.16}$ | $62.56_{\pm1.27}$ | $60.93_{\pm1.79}$ | $\underline{72.75}_{\pm0.84}$ | $87.77_{\pm0.43}$ | $62.42_{\pm1.27}$ | $62.49_{\pm2.29}$ | $68.60_{\pm2.69}$ |
| | NCAD | $55.02_{\pm0.15}$ | $61.71_{\pm0.89}$ | $60.05_{\pm2.03}$ | $64.39_{\pm0.72}$ | $64.30_{\pm1.13}$ | $50.07_{\pm1.88}$ | $45.93_{\pm0.69}$ | $69.80_{\pm1.02}$ |
| | DRA | $54.38_{\pm0.82}$ | $57.26_{\pm1.52}$ | $60.29_{\pm2.37}$ | $59.31_{\pm2.50}$ | $\underline{88.73}_{\pm0.27}$ | $40.57_{\pm2.79}$ | $63.33_{\pm2.45}$ | $72.48_{\pm1.65}$ |
| | MOSAD | $54.76_{\pm0.09}$ | $57.84_{\pm0.25}$ | $\underline{61.58}_{\pm0.18}$ | $71.20_{\pm0.38}$ | $80.42_{\pm1.56}$ | $53.14_{\pm0.87}$ | $62.74_{\pm1.72}$ | $69.80_{\pm2.83}$ |
| | InvAD | $\underline{55.64}_{\pm0.71}$ | $61.32_{\pm5.44}$ | $61.34_{\pm0.88}$ | $71.24_{\pm0.86}$ | $75.71_{\pm0.43}$ | $52.88_{\pm1.71}$ | $50.20_{\pm3.15}$ | $61.63_{\pm4.27}$ |
| | WSAD-DT | $51.47_{\pm0.48}$ | $\underline{63.76}_{\pm1.22}$ | $60.76_{\pm1.58}$ | $70.60_{\pm0.80}$ | $85.18_{\pm1.22}$ | $\underline{64.42}_{\pm1.30}$ | $\underline{65.83}_{\pm3.04}$ | $\underline{78.70}_{\pm0.98}$ |
| | IMPACT | $\mathbf{59.21}_{\pm1.34}$ | $\mathbf{65.76}_{\pm2.17}$ | $\mathbf{64.24}_{\pm1.67}$ | $\mathbf{75.97}_{\pm3.03}$ | $\mathbf{91.96}_{\pm2.32}$ | $\mathbf{68.13}_{\pm2.59}$ | $\mathbf{66.99}_{\pm1.95}$ | $\mathbf{82.91}_{\pm0.74}$ |

# 4. Experiments

## 4.1. Experimental Setup

**Datasets.** To assess the effectiveness of our method, comprehensive experiments are conducted across eight datasets, including four datasets with single anomaly class (UCR, ASD, PSM, SMD) and four datasets with multiple anomaly classes (CT, SAD, PTBXL, TUSZ). These datasets are commonly used in recent literature (Xu et al., 2024; Lai et al., 2024; Liu et al., 2025; Shentu et al., 2025). To verify the robustness, the training data of each dataset was contaminated by 2% anomalies. Details are included in Appendix E.

**Baselines.** We compare our method against state-of-the-art baselines under two primary experimental settings: *unsupervised setting* and *open-set setting*. For *unsupervised setting*, where assuming only normal samples are available in the training phase, seven unsupervised methods are included: TCN-AE (Garg et al., 2021), THOC (Shen et al., 2020), TranAD (Tuli et al., 2022), DCdetector (Yang et al., 2023), GPT4TS (Zhou et al., 2023), COUTA (Xu et al., 2024), and DADA (Shentu et al., 2025). For *open-set setting*, we follow the previous baselines (Ding et al., 2022; Lai et al., 2024) to adopt two protocols for sampling, including general setting and hard setting. The general setting simulates a scenario where labeled anomalies are randomly sampled from all possible anomaly classes, while the hard setting samples labeled anomalies from a single class to evaluate generalization to unseen anomaly classes. The datasets CT, SAD, PTBXL and TUSZ are utilized for the hard setting, as they contain multiple distinct anomaly classes. Given the rarity of anomalies, both settings utilize only a limited number of labeled anomaly samples, fixed at ten per class. Meanwhile, our method is compared with seven closely related baselines, including DevNet (Pang et al., 2019), DSAD (Ruff et al., 2020), NCAD (Carmona et al., 2022), DRA (Ding et al., 2022), MOSAD (Lai et al., 2024), InvAD (Liu et al., 2025), and WSAD-DT (Durani et al., 2025). DevNet, DRA, MOSAD, InvAD and WSAD-DT are specifically designed for open-set environment, while DSAD and NCAD are supervised anomaly detectors.

**Evaluation Metrics.** The performance across all methods and settings is evaluated using the Area Under the Receiver Operating Characteristic Curve (AUC), a widely adopted metric for anomaly detection tasks. All reported results are averaged over five independent runs.

## 4.2. Experimental Results

**General Setting.** Table 1 illustrates the detection performance under both unsupervised and open-set general settings. Our method consistently outperforms all state-of-the-art approaches in AUC across the eight datasets from diverse application domains. Our method averagely obtains 6.37%-44.52% AUC improvement over all contenders. Particularly impressive is our method's performance on complex real-world datasets like CT (91.96%) and TUSZ (82.91%), where it substantially outperforms the second-best methods by large margins. While access to limited seen anomalies during training offers open-set methods an advantage over unsupervised baselines, this same factor causes detectors tend to overfit the limited seen anomalies and lack the ability to resist anomaly contamination. Notably, the superiority of our method demonstrates that our test-risk-driven influence scoring and label flipping strategies can effectively eliminate the impact of potential contamination and further expand the utilization of the known abnormal knowledge.

*Table 2.* Detection accuracy (AUC $\pm$ standard deviation) of IMPACT and its competing methods under the open-set hard settings. All results are in %. The best ones are **boldfaced** and the second best are underlined.

| Dataset | seen anomaly | DevNet | DSAD | NCAD | DRA | MOSAD | InvAD | WSAD-DT | IMPACT |
|---|---|---|---|---|---|---|---|---|---|
| CT | Letter-G | $49.54_{\pm1.03}$ | $73.56_{\pm3.41}$ | $63.27_{\pm0.65}$ | $68.79_{\pm3.43}$ | $\underline{75.64}_{\pm1.50}$ | $75.43_{\pm0.52}$ | $68.83_{\pm1.77}$ | $\mathbf{80.90}_{\pm1.03}$ |
| | Letter-M | $50.75_{\pm1.52}$ | $\underline{75.67}_{\pm1.41}$ | $64.28_{\pm0.58}$ | $74.93_{\pm2.22}$ | $75.15_{\pm0.69}$ | $75.60_{\pm0.89}$ | $72.51_{\pm2.35}$ | $\mathbf{81.65}_{\pm0.38}$ |
| | Letter-Q | $50.91_{\pm1.68}$ | $74.46_{\pm1.16}$ | $64.52_{\pm0.77}$ | $71.85_{\pm3.11}$ | $\underline{76.37}_{\pm0.84}$ | $75.01_{\pm0.77}$ | $74.01_{\pm2.17}$ | $\mathbf{82.50}_{\pm0.37}$ |
| | Letter-W | $51.40_{\pm3.83}$ | $74.23_{\pm3.22}$ | $63.71_{\pm0.39}$ | $73.02_{\pm1.57}$ | $75.75_{\pm0.48}$ | $75.74_{\pm0.62}$ | $\underline{76.20}_{\pm1.07}$ | $\mathbf{80.87}_{\pm1.05}$ |
| | Letter-Z | $49.54_{\pm1.05}$ | $\underline{79.46}_{\pm2.10}$ | $64.22_{\pm0.35}$ | $75.11_{\pm1.38}$ | $78.70_{\pm0.81}$ | $75.33_{\pm0.67}$ | $74.94_{\pm1.36}$ | $\mathbf{83.58}_{\pm0.97}$ |
| | Mean | $50.43_{\pm0.76}$ | $75.48_{\pm2.11}$ | $64.00_{\pm0.45}$ | $72.74_{\pm2.32}$ | $\underline{76.32}_{\pm1.25}$ | $75.42_{\pm0.25}$ | $73.30_{\pm2.54}$ | $\mathbf{81.90}_{\pm1.03}$ |
| SAD | Digit-Zero | $50.41_{\pm1.42}$ | $57.49_{\pm2.01}$ | $45.84_{\pm1.91}$ | $48.04_{\pm2.02}$ | $52.95_{\pm0.72}$ | $50.66_{\pm2.99}$ | $\underline{60.53}_{\pm2.06}$ | $\mathbf{62.85}_{\pm4.79}$ |
| | Digit-Three | $50.00_{\pm0.00}$ | $60.07_{\pm3.61}$ | $47.69_{\pm1.40}$ | $46.90_{\pm1.94}$ | $52.94_{\pm0.28}$ | $49.84_{\pm1.88}$ | $\mathbf{70.26}_{\pm1.17}$ | $\underline{67.34}_{\pm3.66}$ |
| | Digit-Seven | $50.00_{\pm0.00}$ | $60.90_{\pm2.01}$ | $47.06_{\pm1.58}$ | $41.87_{\pm1.92}$ | $52.68_{\pm0.96}$ | $49.93_{\pm2.39}$ | $63.44_{\pm1.09}$ | $\mathbf{70.71}_{\pm2.02}$ |
| | Digit-Eight | $50.00_{\pm0.00}$ | $60.82_{\pm2.87}$ | $47.57_{\pm0.80}$ | $46.77_{\pm3.15}$ | $53.16_{\pm0.88}$ | $51.44_{\pm1.41}$ | $62.36_{\pm1.87}$ | $\mathbf{63.86}_{\pm1.05}$ |
| | Mean | $50.10_{\pm0.18}$ | $59.82_{\pm1.38}$ | $47.04_{\pm0.73}$ | $45.90_{\pm2.38}$ | $52.93_{\pm0.17}$ | $50.47_{\pm0.65}$ | $\underline{64.15}_{\pm3.68}$ | $\mathbf{66.19}_{\pm3.10}$ |
| PTBXL | MI | $50.00_{\pm0.00}$ | $58.69_{\pm2.46}$ | $46.32_{\pm0.64}$ | $60.33_{\pm2.84}$ | $59.61_{\pm4.73}$ | $52.95_{\pm4.88}$ | $\underline{65.27}_{\pm1.45}$ | $\mathbf{65.80}_{\pm3.95}$ |
| | STTC | $50.00_{\pm0.00}$ | $50.60_{\pm4.09}$ | $46.51_{\pm1.29}$ | $51.68_{\pm2.15}$ | $53.36_{\pm4.81}$ | $54.00_{\pm5.18}$ | $\underline{56.42}_{\pm2.61}$ | $\mathbf{59.35}_{\pm3.89}$ |
| | CD | $50.00_{\pm0.00}$ | $54.20_{\pm3.53}$ | $46.25_{\pm0.91}$ | $52.36_{\pm2.87}$ | $52.72_{\pm3.63}$ | $51.44_{\pm7.09}$ | $\mathbf{64.51}_{\pm0.82}$ | $\underline{61.73}_{\pm5.17}$ |
| | HYP | $50.00_{\pm0.00}$ | $58.32_{\pm3.78}$ | $46.11_{\pm0.76}$ | $\underline{60.32}_{\pm1.67}$ | $60.17_{\pm2.38}$ | $52.76_{\pm6.70}$ | $60.02_{\pm2.48}$ | $\mathbf{63.02}_{\pm3.32}$ |
| | Mean | $50.00_{\pm0.00}$ | $55.45_{\pm3.31}$ | $46.30_{\pm0.14}$ | $56.17_{\pm4.16}$ | $56.47_{\pm3.44}$ | $52.79_{\pm0.91}$ | $\underline{61.56}_{\pm3.58}$ | $\mathbf{62.48}_{\pm2.33}$ |
| TUSZ | FNSZ | $49.62_{\pm0.67}$ | $\underline{75.00}_{\pm0.77}$ | $69.94_{\pm2.28}$ | $63.61_{\pm2.01}$ | $69.87_{\pm2.75}$ | $57.15_{\pm8.60}$ | $64.79_{\pm2.00}$ | $\mathbf{76.27}_{\pm1.65}$ |
| | GNSZ | $50.04_{\pm0.06}$ | $53.64_{\pm2.36}$ | $\underline{69.59}_{\pm2.29}$ | $53.00_{\pm3.35}$ | $66.65_{\pm2.15}$ | $54.99_{\pm8.50}$ | $58.60_{\pm1.71}$ | $\mathbf{74.53}_{\pm2.27}$ |
| | CPSZ | $50.04_{\pm0.06}$ | $\underline{72.54}_{\pm1.22}$ | $69.54_{\pm2.30}$ | $56.62_{\pm3.82}$ | $69.71_{\pm2.49}$ | $56.04_{\pm8.98}$ | $56.65_{\pm1.72}$ | $\mathbf{76.30}_{\pm3.17}$ |
| | Mean | $49.90_{\pm0.20}$ | $67.06_{\pm9.54}$ | $\underline{69.69}_{\pm0.18}$ | $57.74_{\pm4.40}$ | $68.74_{\pm1.48}$ | $56.06_{\pm0.88}$ | $60.01_{\pm3.47}$ | $\mathbf{75.70}_{\pm0.83}$ |

**Hard Setting.** Table 2 shows the performance comparison under the hard setting. Our method demonstrates remarkable generalization capability, averagely achieving 10.52%-42.83% AUC improvement over seven competitors. The superior performance underscores the effectiveness of our unseen abnormality learning. By leveraging influence scores to identify samples for feature perturbation and incorporating a dedicated unseen abnormality learning head, our method successfully bridges the gap between seen and unseen anomalies, achieving robust detection performance even when faced with completely novel anomaly classes.

### 4.3. Ablation Study

This experiment first validates the effectiveness of label flipping, the importance of each abnormality learning, and the role of feature deviation in anomaly scoring. Based on AUC results reported in Table 3, It is worth noting that removing the label-flipped contaminated samples (**w/o** $\mathcal{D}'_{con}$) consistently leads to performance degradation across all datasets, which indicates that correcting contaminated samples is crucial for enhancing model robustness. The absence of seen abnormality learning (**w/o** $L_{seen}$) results in the most significant performance drop, particularly in the general setting (*e.g.*, TUSZ declines from 82.91% to 70.91%), underscoring that utilizing known anomaly patterns can substantially enhance discriminative power. While removing unseen abnormality learning (**w/o** $L_{unseen}$) shows less impact in the general setting as all anomaly classes are provided, its omission severely harms performance under the hard setting (*e.g.*,

TUSZ drops to 60.92%), confirming that capturing unknown abnormality is essential when anomalies exhibit high diversity. Lastly, ablating the feature deviation score (**w/o** $s_f$) leads to noticeable deterioration, validating that feature deviation provides complementary information beyond the abnormality score.

To further validate that the improvements stem from the influence-guided design, we compare against three randomized alternatives and one variant that directly uses unflipped contaminated samples. Replacing the influence-guided reference set with a randomly selected one (**w/** rnd $\mathcal{D}_{ref}$) causes substantial performance degradation across all datasets, which demonstrates that the quality of $\mathcal{D}_{ref}$ is critical. Similarly, replacing influence-guided label flipping with random label flipping (**w/** rnd $\mathcal{D}'_{con}$) consistently degrades performance, as random flipping indiscriminately corrupts genuinely normal samples alongside true contaminants, introducing additional label noise. Replacing influence-guided feature perturbation with random Gaussian perturbation (**w/** rnd $\mathcal{W}'_{per}$) also leads to consistent drops (*e.g.*, TUSZ declines from 75.70% to 62.87% in the hard setting), confirming that random perturbations fail to target risk-increasing directions in the feature space and thus produce low-quality pseudo anomalies that do not effectively simulate unseen anomaly distributions. Finally, directly incorporating the original unflipped contaminated samples (**w/** $\mathcal{D}_{con}$) demonstrates that contamination severely misleads the model. Collectively, these results confirm that the influence-guided design is indispensable, and naive random alternatives consistently fail to replicate its benefits.

*Table 3.* Ablation study results (as %) of ours and its variants under both general and hard settings. The best results are **boldfaced**.

| Dataset | Ours | w/o $\mathcal{D}'_{con}$ | w/o $L_{seen}$ | w/o $L_{unseen}$ | w/o $s_f$ | w/ $\mathcal{D}_{con}$ | w/ rnd $\mathcal{D}_{ref}$ | w/ rnd $\mathcal{D}'_{con}$ | w/ rnd $\mathcal{W}'_{per}$ |
|---|---|---|---|---|---|---|---|---|---|
| **General Setting** | | | | | | | | | |
| UCR | $59.21_{\pm1.34}$ | $58.95_{\pm1.86}$ | $50.14_{\pm4.26}$ | $\mathbf{59.27}_{\pm1.55}$ | $54.25_{\pm2.24}$ | $55.74_{\pm1.21}$ | $52.12_{\pm3.14}$ | $53.47_{\pm4.22}$ | $53.63_{\pm3.05}$ |
| ASD | $\mathbf{65.76}_{\pm2.17}$ | $64.34_{\pm2.97}$ | $57.39_{\pm3.25}$ | $65.60_{\pm2.73}$ | $64.67_{\pm4.13}$ | $60.23_{\pm1.44}$ | $58.43_{\pm3.56}$ | $57.89_{\pm2.54}$ | $58.14_{\pm2.41}$ |
| PSM | $\mathbf{64.24}_{\pm1.67}$ | $63.37_{\pm2.37}$ | $61.67_{\pm2.49}$ | $63.47_{\pm2.35}$ | $62.38_{\pm3.43}$ | $62.44_{\pm0.96}$ | $60.13_{\pm3.76}$ | $59.82_{\pm2.35}$ | $60.47_{\pm1.97}$ |
| SMD | $\mathbf{75.97}_{\pm3.03}$ | $74.07_{\pm2.80}$ | $72.28_{\pm2.24}$ | $75.63_{\pm1.39}$ | $69.02_{\pm3.34}$ | $73.13_{\pm1.25}$ | $67.49_{\pm3.44}$ | $69.34_{\pm1.31}$ | $71.82_{\pm2.22}$ |
| CT | $91.96_{\pm2.32}$ | $91.81_{\pm2.05}$ | $90.24_{\pm2.78}$ | $\mathbf{92.16}_{\pm1.84}$ | $91.32_{\pm2.12}$ | $90.42_{\pm1.51}$ | $87.75_{\pm2.76}$ | $86.13_{\pm1.36}$ | $87.54_{\pm2.41}$ |
| SAD | $\mathbf{68.13}_{\pm2.59}$ | $67.56_{\pm3.62}$ | $66.88_{\pm3.34}$ | $67.95_{\pm4.29}$ | $68.04_{\pm3.62}$ | $66.94_{\pm2.07}$ | $65.43_{\pm2.86}$ | $62.27_{\pm1.37}$ | $63.89_{\pm2.26}$ |
| PTBXL | $66.99_{\pm1.95}$ | $65.49_{\pm2.39}$ | $63.41_{\pm2.68}$ | $\mathbf{67.11}_{\pm2.20}$ | $65.13_{\pm1.93}$ | $63.55_{\pm1.46}$ | $62.31_{\pm3.77}$ | $61.58_{\pm1.09}$ | $62.12_{\pm2.46}$ |
| TUSZ | $\mathbf{82.91}_{\pm0.74}$ | $82.62_{\pm0.88}$ | $70.91_{\pm1.77}$ | $82.63_{\pm0.87}$ | $74.18_{\pm6.47}$ | $71.37_{\pm3.53}$ | $69.76_{\pm2.76}$ | $72.43_{\pm1.47}$ | $74.68_{\pm2.58}$ |
| **Hard Setting** | | | | | | | | | |
| CT(mean) | $\mathbf{81.90}_{\pm1.03}$ | $78.95_{\pm1.29}$ | $77.32_{\pm2.08}$ | $76.34_{\pm1.39}$ | $79.56_{\pm1.53}$ | $77.68_{\pm1.74}$ | $74.22_{\pm3.11}$ | $76.23_{\pm2.31}$ | $77.12_{\pm2.11}$ |
| SAD(mean) | $\mathbf{66.19}_{\pm3.10}$ | $65.66_{\pm3.04}$ | $64.32_{\pm2.87}$ | $60.09_{\pm2.73}$ | $63.60_{\pm1.55}$ | $64.93_{\pm2.16}$ | $59.74_{\pm2.93}$ | $63.23_{\pm1.32}$ | $63.46_{\pm3.31}$ |
| PTBXL(mean) | $\mathbf{62.48}_{\pm2.33}$ | $61.16_{\pm2.37}$ | $59.37_{\pm2.16}$ | $53.00_{\pm1.44}$ | $56.93_{\pm2.43}$ | $60.23_{\pm2.46}$ | $51.56_{\pm2.13}$ | $56.88_{\pm2.75}$ | $56.41_{\pm2.43}$ |
| TUSZ(mean) | $\mathbf{75.70}_{\pm0.83}$ | $72.64_{\pm1.55}$ | $70.11_{\pm0.78}$ | $60.92_{\pm1.27}$ | $66.61_{\pm0.72}$ | $70.84_{\pm1.21}$ | $59.78_{\pm1.26}$ | $63.57_{\pm1.68}$ | $62.87_{\pm1.74}$ |

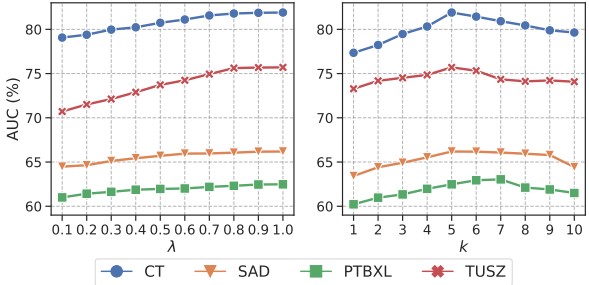

*Figure 3.* Sensitivity analysis results (AUC performance w.r.t. different hyperparameters) under the hard setting.

## 4.4. Sensitivity Analysis

To further investigate the impact of unseen abnormality learning, we conduct an analysis of two key hyperparameters in IMPACT under the hard setting, the weight adapting coefficient ($\lambda$) and the number of perturbed samples ($k$). The results are shown in Figure 3. We observe that the detection performance is improved and tends to be stable with the increase of $\lambda$, which indicates the introduced unseen abnormality learning can contribute to stronger generalization under challenging open-set conditions. Notably, increasing $k$ does not always result in improved performance. This is mainly because that only a fraction of normal samples in each batch exhibit transformation compatibility for generating unseen anomalies. When an excessive number of normal samples undergo perturbation, the intrinsic normality within certain samples would be forcibly corrupted, consequently inducing learning biases during model optimization. The results indicate that $\lambda = 1.0$ and $k = 5$ are sufficient for effective anomaly generalization, which are the default settings for IMPACT throughout comprehensive experiments.

## 5. Conclusion

We propose IMPACT, a robust framework for open-set TSAD that effectively tackles the dual challenges of anomaly contamination and generation to unseen anomalies. By integrating influence functions into the anomaly detection pipeline, we repurpose contaminated samples into valuable supervised knowledge through label flipping. Furthermore, we propose feature perturbation to generate high-quality pseudo anomalies, thereby effectively detecting unseen anomaly patterns. Extensive experiments demonstrate that IMPACT outperforms existing state-of-the-art methods.

## Acknowledgements

This work was supported in part by the National Science and Technology Major Project (Grant No.2022ZD0115302), the National Natural Science Foundation of China (Grant No.61379052, 62406328, and 62506369), the Science Foundation of Ministry of Education of China (Grant No.2018A02002), and the Natural Science Foundation for Distinguished Young Scholars of Hunan Province (Grant No.14JJ1026). G. Pang's participation in this research was partially supported by the National Research Foundation, Singapore and CyberSG R&D Programme Office under its Translation and Innovation Grants (CRPO-GC4-SMU-001), the Singapore Ministry of Education (MOE) Academic Research Fund (AcRF) Tier 1 Grant (24-SIS-SMU-008), A*STAR under its MTC YIRG Grant (M24N8c0103), and the Lee Kong Chian Fellowship (T050273).

## Impact Statement

This paper presents work whose goal is to advance the field of Machine Learning. There are many potential societal consequences of our work, none which we feel must be specifically highlighted here.

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

## A. Proof of Theorem 1

*Proof.* Recall the output multi-channel anomaly scores $f(\boldsymbol{x}_i, \theta) = \boldsymbol{s} \in \mathbb{R}^r$. We first do not assume a specific distribution, and let $\boldsymbol{s}$ follow a latent distribution $S$ with covariance $\Sigma$, probability density function $p(\boldsymbol{s})$. Then we know $p(\boldsymbol{s})$ is the density satisfying $\int s_i s_j p(\boldsymbol{s}) d\boldsymbol{s} = \Sigma_{ij}$ for all $i, j$. $\mathcal{H}(S) = \mathbb{E}[-\log p(S)] = -\int_S p(\boldsymbol{s}) \log p(\boldsymbol{s}) d\boldsymbol{s}$. Let $\psi_\Sigma$ be the density of distribution $\mathcal{N}(\boldsymbol{\mu}, \Sigma)$:

$$\psi_\Sigma(\boldsymbol{s}) = \frac{1}{\left(\sqrt{2\pi}\right)^r |\Sigma|^{\frac{1}{2}}} e^{-\frac{1}{2}(\boldsymbol{s}-\boldsymbol{\mu})^T \Sigma^{-1}(\boldsymbol{s}-\boldsymbol{\mu})}. \tag{22}$$

The expectation $\mathbb{E}[s_i s_j]$ is given by: $\mathbb{E}[s_i s_j] = \Sigma_{ij} + \mu_i \mu_j$. This is because the covariance between $s_i$ and $s_j$ is $\Sigma_{ij}$, and the expectation of the product is the sum of the covariance and the product of the means $\mu_i \mu_j$. When the distribution is centered (*i.e.*, $\boldsymbol{\mu} = 0$), the expectation simplifies to: $\mathbb{E}[s_i s_j] = \int s_i s_j \psi_\Sigma(\boldsymbol{s}) d\boldsymbol{s} = \Sigma_{ij}$. Then we have:

$$\int s_i s_j p(\boldsymbol{s}) d\boldsymbol{s} = \int s_i s_j \psi_\Sigma(\boldsymbol{s}) d\boldsymbol{s}, \tag{23}$$

which means both $p(\boldsymbol{s})$ and $\psi_\Sigma(\boldsymbol{s})$ yield the same quadratic moments. Since we know the KL-divergence $D_{\text{KL}}(p||q)$ between two densities $p$ and $q$ is defined by: $D_{\text{KL}}(p||q) = \int p \log \frac{q}{p}$, we have the fact:

$$\begin{aligned}
-D_{\text{KL}}(p||q) &= \int p \log \frac{q}{p} \\
&\leq \log \int p \frac{q}{p} \qquad \text{(by Jensen's inequality)} \\
&= \log \int q \leq \log 1 = 0.
\end{aligned} \tag{24}$$

KL-divergence achieves 0 iff we have equality in Jensen's inequality, which occurs iff $p = q$. Then we have:

$$\begin{aligned}
0 &\leq D_{\text{KL}}(p||\psi_\Sigma) \\
&= \int_S p \log(p/\psi_\Sigma) d\boldsymbol{s} \\
&= \int_S p \log p \, d\boldsymbol{s} - \int_S p \log \psi_\Sigma d\boldsymbol{s} \\
&= -\mathcal{H}(S) - \int_S p \log \psi_\Sigma d\boldsymbol{s} \\
&= -\mathcal{H}(S) - \int_S \psi_\Sigma \log \psi_\Sigma d\boldsymbol{s} \\
&= -\mathcal{H}(S) + \mathcal{H}(\psi_\Sigma),
\end{aligned} \tag{25}$$

where the substitution $\int_S p \log \psi_\Sigma \mathrm{d}s = \int_S \psi_\Sigma \log \psi_\Sigma \mathrm{d}s$ follows from the fact that $p$ and $\psi_\Sigma$ yield the same moments of the quadratic form $\log \psi_\Sigma$, as indicated in Equation (23). Therefore, $\mathcal{H}(S) \le \mathcal{H}(\psi_\Sigma)$. Below we further compute $\mathcal{H}(\psi_\Sigma)$:

$$
\begin{aligned}
\mathcal{H}(\psi_\Sigma) &= -\int_S \psi_\Sigma(s) \left[ -\frac{1}{2}(s-\mu)^T \Sigma^{-1}(s-\mu) - \ln\left(\sqrt{2\pi}\right)^r |\Sigma|^{\frac{1}{2}} \right] \mathrm{d}s \\
&= \frac{1}{2} \mathbb{E}\left[ \sum_{i,j}(s_i - \mu_i)\left(\Sigma^{-1}\right)_{ij}(s_j - \mu_j) \right] + \frac{1}{2}\ln(2\pi)^r|\Sigma| \\
&= \frac{1}{2} \mathbb{E}\left[ \sum_{i,j}(s_i - \mu_i)(s_j - \mu_j)\left(\Sigma^{-1}\right)_{ij} \right] + \frac{1}{2}\ln(2\pi)^r|\Sigma| \\
&= \frac{1}{2} \sum_{i,j} \mathbb{E}[(s_j - \mu_j)(s_i - \mu_i)]\left(\Sigma^{-1}\right)_{ij} + \frac{1}{2}\ln(2\pi)^r|\Sigma| \\
&= \frac{1}{2} \sum_j \sum_i \Sigma_{ji}\left(\Sigma^{-1}\right)_{ij} + \frac{1}{2}\ln(2\pi)^r|\Sigma| \\
&= \frac{1}{2} \sum_j (\Sigma\Sigma^{-1})_{jj} + \frac{1}{2}\ln(2\pi)^r|\Sigma| \\
&= \frac{1}{2} \sum_j I_{jj} + \frac{1}{2}\ln(2\pi)^r|\Sigma| \\
&= \frac{r}{2} + \frac{1}{2}\ln(2\pi)^r|\Sigma| \\
&= \frac{1}{2}\ln(2\pi e)^r|\Sigma| \quad \text{nats} \\
&= \frac{1}{2}\log(2\pi e)^r \det \Sigma \quad \text{bits}.
\end{aligned}
\tag{26}
$$

Then we have the following bound on entropy:

$$
\mathcal{H}(S) \le \mathcal{H}(\psi_\Sigma) = \frac{1}{2}\log((2\pi e)^r \det \Sigma),
\tag{27}
$$

which holds with equality iff $S$ is jointly Gaussian.

Assuming the latent distribution $S$ follows an isotropic Gaussian, $S \sim \mathcal{N}(\mu, \sigma^2 I)$, we have:

$$
\mathcal{H}(S) = \frac{1}{2}\log((2\pi e)^r \det \sigma^2 I) = \frac{1}{2}\log((2\pi e\sigma^2)^r \cdot 1) = \frac{r}{2}(1 + \log(2\pi\sigma^2)) \propto \log\sigma^2.
\tag{28}
$$

$\square$

## B. Proof of Theorem 2

*Proof.* It is worth noting that $\hat{\theta}_{\epsilon_i 1} = \arg\min_{\theta \in \Theta} \frac{1}{N}\sum_{n=1}^N L(z_n, \theta) + \epsilon_i L(z_{i1}, \theta) - \epsilon_i L(z_i, \theta)$. In this way, label flipping on the contaminated training sample $z_i$ in $\mathcal{D}_{con}$ means setting $\epsilon_i = \frac{1}{N}$. According to Equations (11) and (12), we have:

$$
\begin{aligned}
L(\mathcal{V}, \hat{\theta}_{\epsilon_1}) - L(\mathcal{V}, \hat{\theta}) &= \frac{1}{|\mathcal{D}_{con}|} \sum_{z_i \in \mathcal{D}_{con}} L(\mathcal{V}, \hat{\theta}_{\epsilon_{i1}}) - L(\mathcal{V}, \hat{\theta}) \\
&\approx \frac{1}{|\mathcal{D}_{con}|} \sum_{z_i \in \mathcal{D}_{con}} \epsilon_i \times \sum_{z_t \in \mathcal{V}} \mathcal{I}_{L1}(z_i, z_t) \\
&= \frac{1}{N \cdot |\mathcal{D}_{con}|} \sum_{z_i \in \mathcal{D}_{con}} \sum_{z_t \in \mathcal{V}} -2\mathcal{I}_L(z_i, z_t) \\
&= \frac{-2}{N \cdot |\mathcal{D}_{con}|} \sum_{z_i \in \mathcal{D}_{con}} \mathcal{I}_L(z_i) < 0.
\end{aligned}
\tag{29}
$$

$\square$

## C. Proof of Theorem 3

*Proof.* According to Equation (15), we first formalize $\mathcal{I}_{per}(\boldsymbol{w}_i)$ as follows:

$$
\begin{aligned}
\mathcal{I}_{per}(\boldsymbol{w}_i) &= -\nabla_{\theta_h} L(\mathcal{V}, \hat{\theta}_h)^\top H_{\hat{\theta}_h}^{-1} [\nabla_{\boldsymbol{\varphi}} \nabla_{\theta_h} L(\boldsymbol{w}_i, \hat{\theta}_h)] \\
&= \sum_{\boldsymbol{w}_t \in \mathcal{V}} -\nabla_{\theta_h} L(\boldsymbol{w}_t, \hat{\theta}_h)^\top H_{\hat{\theta}_h}^{-1} [\nabla_{\boldsymbol{\varphi}} \nabla_{\theta_h} L(\boldsymbol{w}_i, \hat{\theta}_h)],
\end{aligned} \tag{30}
$$

where $\theta_h$ is flatten into $\mathbb{R}^{m \times 1}$, $\nabla_{\theta_h} L(\boldsymbol{w}_t, \hat{\theta}_h)^\top \in \mathbb{R}^{1 \times m}$, $H_{\hat{\theta}_h}^{-1} \in \mathbb{R}^{m \times m}$, $\nabla_{\boldsymbol{\varphi}} \nabla_{\theta_h} L(\boldsymbol{w}_i, \hat{\theta}_h) \in \mathbb{R}^{m \times d}$, and $\mathcal{I}_{per}(\boldsymbol{w}_i) \in \mathbb{R}^{1 \times d}$. Consider the embedded features $\boldsymbol{\varphi}_i \in \mathbb{R}^{d \times 1}$, and let $\boldsymbol{\varphi}_{i\zeta_i} = \boldsymbol{\varphi}_i + \alpha \mathcal{I}_{per}(\boldsymbol{w}_i)^\top = (I + \alpha P)\boldsymbol{\varphi}_i$, then we can obtain $\left| \frac{d\boldsymbol{\varphi}_{i\zeta_i}}{d\boldsymbol{\varphi}_i} \right| = |(I + \alpha P)|$. Assuming that the original features $\boldsymbol{\varphi}_i$ are i.i.d. drawn from the distribution with the probability density function $p(\boldsymbol{\varphi}_i)$, then we can conduct the perturbed features $\boldsymbol{\varphi}_{i\zeta_i}$ are i.i.d. drawn from the distribution with $p(\boldsymbol{\varphi}_{i\zeta_i}) = p(\boldsymbol{\varphi}_i) \left| \frac{d\boldsymbol{\varphi}_i}{d\boldsymbol{\varphi}_{i\zeta_i}} \right| = p(\boldsymbol{\varphi}_i) \left| (I + \alpha P)^{-1} \right| = p(\boldsymbol{\varphi}_i) |I + \alpha P|^{-1}$.

By using the KL-divergence to measure the discrepancy between the original feature distribution and the perturbed feature distribution, we have

$$
D_{\mathrm{KL}}(p(\boldsymbol{\varphi}_i) \| p(\boldsymbol{\varphi}_{i\zeta_i})) = \mathbb{E}_{p(\boldsymbol{\varphi}_i)} \log \frac{p(\boldsymbol{\varphi}_i)}{p(\boldsymbol{\varphi}_{i\zeta_i})} = |\log|I + \alpha P||. \tag{31}
$$

Recall $P\boldsymbol{\varphi}_i = \mathcal{I}_{per}(\boldsymbol{w}_i)^\top$, we can further let $P = \mathcal{I}_{per}(\boldsymbol{w}_i)^\top \cdot \frac{\boldsymbol{\varphi}_i^\top}{\|\boldsymbol{\varphi}_i\|^2}$, which satisfies

$$
P\boldsymbol{\varphi}_i = \mathcal{I}_{per}(\boldsymbol{w}_i)^\top \cdot \frac{\boldsymbol{\varphi}_i^\top}{\|\boldsymbol{\varphi}_i\|^2} \cdot \boldsymbol{\varphi}_i = \mathcal{I}_{per}(\boldsymbol{w}_i)^\top \cdot \frac{\boldsymbol{\varphi}_i^\top \boldsymbol{\varphi}_i}{\|\boldsymbol{\varphi}_i\|^2} = \mathcal{I}_{per}(\boldsymbol{w}_i)^\top. \tag{32}
$$

Since $P$ is the matrix of rank 1, we can know there are only one nonzero eigenvalue and remaining $d - 1$ zero eigenvalue. To calculate $|\log|I + \alpha P||$, below we first derive that the only one nonzero eigenvalue of $P$ is $\lambda = \frac{\mathcal{I}_{per}(\boldsymbol{w}_i)^\top \boldsymbol{\varphi}_i}{\|\boldsymbol{\varphi}_i\|^2}$.

For any $\boldsymbol{x} \in \mathbb{R}^{d \times 1}$, we have:

$$
P\boldsymbol{x} = \frac{\mathcal{I}_{per}(\boldsymbol{w}_i)^\top \boldsymbol{\varphi}_i^\top \boldsymbol{x}}{\|\boldsymbol{\varphi}_i\|^2} = \frac{\boldsymbol{\varphi}_i^\top \boldsymbol{x}}{\|\boldsymbol{\varphi}_i\|^2} \mathcal{I}_{per}(\boldsymbol{w}_i)^\top. \tag{33}
$$

Therefore, $P$ maps any vectors to the direction of $\mathcal{I}_{per}(\boldsymbol{w}_i)^\top$. Let $\boldsymbol{x} = \mathcal{I}_{per}(\boldsymbol{w}_i)^\top$, we have:

$$
P\mathcal{I}_{per}(\boldsymbol{w}_i)^\top = \frac{\boldsymbol{\varphi}_i^\top \mathcal{I}_{per}(\boldsymbol{w}_i)^\top}{\|\boldsymbol{\varphi}_i\|^2} \mathcal{I}_{per}(\boldsymbol{w}_i)^\top, \tag{34}
$$

where $P$ modifies the magnitude of $\mathcal{I}_{per}(\boldsymbol{w}_i)^\top$ rather than the direction. Thus $\mathcal{I}_{per}(\boldsymbol{w}_i)^\top$ is the eigenvector of $P$, and the corresponding nonzero eigenvalue is $\lambda = \frac{\boldsymbol{\varphi}_i^\top \mathcal{I}_{per}(\boldsymbol{w}_i)^\top}{\|\boldsymbol{\varphi}_i\|^2}$.

Then we have $|I + \alpha P| = (1 + \alpha\lambda) \cdot 1^{d-1} = 1 + \alpha\lambda$, which can be further expanded as:

$$
\begin{aligned}
|I + \alpha P| &= 1 + \alpha \frac{\boldsymbol{\varphi}_i^\top \mathcal{I}_{per}(\boldsymbol{w}_i)^\top}{\|\boldsymbol{\varphi}_i\|^2} \\
&= 1 + \alpha \frac{\boldsymbol{\varphi}_i^\top \sum_{\boldsymbol{w}_t \in \mathcal{V}} -[\nabla_{\boldsymbol{\varphi}} \nabla_{\theta_h} L(\boldsymbol{w}_i, \hat{\theta}_h)]^\top H_{\hat{\theta}_h}^{-1} \nabla_{\theta_h} L(\boldsymbol{w}_t, \hat{\theta}_h)}{\|\boldsymbol{\varphi}_i\|^2}.
\end{aligned} \tag{35}
$$

Similar to Equation (10), we simplify the loss $L(\boldsymbol{w}_i, \theta_h)$ as: $L(\boldsymbol{w}_i, \theta_h) = (1 - y_i)h(\boldsymbol{\varphi}_i, \theta_h) + y_i \max\left(0, a - h(\boldsymbol{\varphi}_i, \theta_h)\right) = h(\boldsymbol{\varphi}_i, \theta_h)$, and treat the learning head as a linear transformation. In this way, $m = d$ and $L(\boldsymbol{w}_i, \theta_h) = \theta_h^\top \boldsymbol{\varphi}_i$, and we obtain $\nabla_{\theta_h} L(\boldsymbol{w}_i, \hat{\theta}_h)^\top = \boldsymbol{\varphi}_i^\top$. Then Equation (35) can be approximated as:

$$
\begin{aligned}
|I + \alpha P| &\approx 1 + \alpha \frac{\nabla_{\theta_h} L(\boldsymbol{w}_i, \hat{\theta}_h)^\top \sum_{\boldsymbol{w}_t \in \mathcal{V}} -H_{\hat{\theta}_h}^{-1} \nabla_{\theta_h} L(\boldsymbol{w}_t, \hat{\theta}_h)}{\|\boldsymbol{\varphi}_i\|^2} \\
&= 1 + \alpha \frac{\sum_{\boldsymbol{w}_t \in \mathcal{V}} -\nabla_{\theta_h} L(\boldsymbol{w}_t, \hat{\theta}_h)^\top H_{\hat{\theta}_h}^{-1} \nabla_{\theta_h} L(\boldsymbol{w}_i, \hat{\theta}_h)}{\|\boldsymbol{\varphi}_i\|^2}.
\end{aligned} \tag{36}
$$

Recall Equation (5), we have

$$
\begin{aligned}
\mathcal{I}_{L\theta_h}(\boldsymbol{w}_i, \boldsymbol{w}_t) &\triangleq \frac{dL(\boldsymbol{w}_t, \hat{\theta}_{h,\epsilon_i})}{d\epsilon_i}\Big|_{\epsilon_i=0} \\
&= -\nabla_{\theta_h} L(\boldsymbol{w}_t, \hat{\theta}_h)^{\top} H_{\hat{\theta}_h}^{-1} \nabla_{\theta_h} L(\boldsymbol{w}_i, \hat{\theta}_h).
\end{aligned}
\tag{37}
$$

Since the perturbed samples are selected from $\mathcal{D}_{per}$ where $\mathcal{I}_{L\theta_h}(\boldsymbol{w}_i) = \sum_{\boldsymbol{w}_t \in \mathcal{V}} \mathcal{I}_{L\theta_h}(\boldsymbol{w}_i, \boldsymbol{w}_t) < 0$, we can define $-\beta$ as the upper bound, where $\beta > 0$. Then Equation (35) has the following upper bound:

$$
\begin{aligned}
|I + \alpha P| &\approx 1 + \alpha \frac{\sum_{\boldsymbol{w}_t \in \mathcal{V}} \mathcal{I}_{L\theta_h}(\boldsymbol{w}_i, \boldsymbol{w}_t)}{\|\boldsymbol{\varphi}_i\|^2} \\
&= 1 + \alpha \frac{\mathcal{I}_{L\theta_h}(\boldsymbol{w}_i)}{\|\boldsymbol{\varphi}_i\|^2} \leq 1 - \frac{\alpha\beta}{\|\boldsymbol{\varphi}_i\|^2} < 1,
\end{aligned}
\tag{38}
$$

where $\alpha, \beta > 0$. Since $\|\boldsymbol{\varphi}_i\|^2 \gg \alpha\beta$, then we have $\frac{\alpha\beta}{\|\boldsymbol{\varphi}_i\|^2} \in (0, 1)$, and Equation (31) has the following lower bound:

$$
\begin{aligned}
D_{\mathrm{KL}}(p(\boldsymbol{\varphi}_i)\|p(\boldsymbol{\varphi}_{i\zeta_i})) &= |\log|I + \alpha P|| \\
&\geq -\log(1 - \frac{\alpha\beta}{\|\boldsymbol{\varphi}_i\|^2}) \\
&= \log(\frac{\|\boldsymbol{\varphi}_i\|^2}{\|\boldsymbol{\varphi}_i\|^2 - \alpha\beta}) \\
&= \log(\frac{1}{1 - \frac{\alpha\beta}{\|\boldsymbol{\varphi}_i\|^2}}) \geq \frac{\alpha\beta}{\|\boldsymbol{\varphi}_i\|^2},
\end{aligned}
\tag{39}
$$

where the last inequality holds because of the fact $\log(\frac{1}{1-x}) \geq x$ for $x \in (0, 1)$.

$\square$

# D. Proof of Theorem 4

*Proof.* Similar to Equations (7) and (12), we denote the optimal parameters as $\hat{\theta}_{h,\epsilon_i\zeta_i\mathbf{1}}$ after perturbing each feature $\boldsymbol{w}_i = (\boldsymbol{\varphi}_i, 0)$ from feature set $\mathcal{W}_{per}$, and quantify the overall influence of $(\boldsymbol{\varphi}_i, 0) \mapsto (\boldsymbol{\varphi}_{i\zeta_i}, 1)$ on the test risk over $\mathcal{V}$ as:

$$
\begin{aligned}
L(\mathcal{V}, \hat{\theta}_{h,\epsilon\zeta\mathbf{1}}) - L(\mathcal{V}, \hat{\theta}_h) &= \frac{1}{|\mathcal{W}_{per}|} \sum_{\boldsymbol{w}_i \in \mathcal{W}_{per}} L(\mathcal{V}, \hat{\theta}_{h,\epsilon_i\zeta_i\mathbf{1}}) - L(\mathcal{V}, \hat{\theta}_h) \\
&= \frac{1}{|\mathcal{W}_{per}|} \sum_{\boldsymbol{w}_i \in \mathcal{W}_{per}} L(\mathcal{V}, \hat{\theta}_{h,\epsilon_i\zeta_i\mathbf{1}}) - L(\mathcal{V}, \hat{\theta}_{h,\epsilon_i\zeta_i}) + L(\mathcal{V}, \hat{\theta}_{h,\epsilon_i\zeta_i}) - L(\mathcal{V}, \hat{\theta}_h) \\
&\approx \frac{1}{|\mathcal{W}_{per}|} \sum_{\boldsymbol{w}_i \in \mathcal{W}_{per}} \epsilon_i \times \sum_{\boldsymbol{w}_t \in \mathcal{V}} -2\mathcal{I}_{L\zeta_i}(\boldsymbol{w}_i, \boldsymbol{w}_t) + \mathcal{I}_{L\zeta_i}(\boldsymbol{w}_i, \boldsymbol{w}_t) \\
&= \frac{1}{|\mathcal{W}_{per}|} \sum_{\boldsymbol{w}_i \in \mathcal{W}_{per}} \epsilon_i \times \sum_{\boldsymbol{w}_t \in \mathcal{V}} -\mathcal{I}_{L\zeta_i}(\boldsymbol{w}_i, \boldsymbol{w}_t).
\end{aligned}
\tag{40}
$$

Similar to Appendix B, we can calculate the gap in test risk before and after the feature perturbation by setting $\epsilon_i = \frac{1}{N}$. Recall that we set feature perturbation $\zeta$ as $\alpha\mathcal{I}_{per}(\boldsymbol{w}_i)^{\top}$ for each feature $\boldsymbol{w}_i$ from $\mathcal{W}_{per}$, then combing Equations (15)

and (40), we have:

$$
\begin{aligned}
L(\mathcal{V}, \hat{\theta}_{h,\epsilon\zeta\mathbf{1}}) - L(\mathcal{V}, \hat{\theta}_h) &\approx \frac{1}{N \cdot |\mathcal{W}_{per}|} \sum_{\boldsymbol{w}_i \in \mathcal{W}_{per}} \sum_{\boldsymbol{w}_t \in \mathcal{V}} -\mathcal{I}_{L\zeta_i}(\boldsymbol{w}_i, \boldsymbol{w}_t) \\
&\approx -\frac{1}{N \cdot |\mathcal{W}_{per}|} \sum_{\boldsymbol{w}_i \in \mathcal{W}_{per}} \mathcal{I}_{per}(\boldsymbol{w}_i)\zeta_i \\
&= -\frac{1}{N \cdot |\mathcal{W}_{per}|} \sum_{\boldsymbol{w}_i \in \mathcal{W}_{per}} \mathcal{I}_{per}(\boldsymbol{w}_i) \times \alpha \mathcal{I}_{per}(\boldsymbol{w}_i)^\top \\
&= -\frac{\alpha}{N \cdot |\mathcal{W}_{per}|} \sum_{\boldsymbol{w}_i \in \mathcal{W}_{per}} \|\mathcal{I}_{per}(\boldsymbol{w}_i)\|_2^2 < 0.
\end{aligned}
\tag{41}
$$

$\square$

## E. Dataset Details

*Table 4.* Details in eight datasets. #lens and #dims indicate the length of a time interval, and the number of variables. #classes represents the number of anomaly classes. #train and #test denote the number of training and test samples. #anom is the number of anomalies over all the test samples.

| Dataset | #lens | #dims | #classes | #train | #test | #anom |
|---------|-------|-------|----------|--------|-------|-------|
| UCR | 100 | 1 | 1 | 54,106 | 70,512 | 172 |
| ASD | 100 | 19 | 1 | 1,043 | 468 | 78 |
| PSM | 100 | 25 | 1 | 1,351 | 798 | 272 |
| SMD | 100 | 38 | 1 | 7,228 | 6,084 | 528 |
| CT | 205 | 3 | 5 | 889 | 1,715 | 412 |
| SAD | 93 | 13 | 4 | 4,040 | 2,200 | 880 |
| PTBXL | 1000 | 12 | 4 | 8,323 | 2,198 | 1,286 |
| TUSZ | 1000 | 19 | 3 | 39,120 | 4,564 | 304 |

Table 4 summarizes the statistical details of these datasets. Below we introduce each dataset in detail.

**UCR** (Wu & Keogh, 2021) contains 250 heterogeneous univariate sub-datasets from various natural sources. Each signal is anomaly-free during training and contains exactly one anomaly in testing. We combined all the training data together and selected a portion of the test anomalies for anomaly contamination injection and anomaly exploitation.

**ASD** (Li et al., 2021b) originates from production environments of a major Internet service provider. It comprises 19 complementary system metrics encompassing CPU utilization, memory allocation, network throughput, and virtual machine performance indicators. Each anomaly in the test set is labeled by domain experts using incident reports, offering reliable ground truth for anomaly detection research.

**PSM** (Abdulaal et al., 2021) is sourced internally from eBay's distributed application infrastructure, comprising operational telemetry collected from multiple server nodes within their production environment. The dataset captures a comprehensive suite of server performance indicators, including CPU utilization, memory allocation and consumption metrics, and so on.

**SMD** (Su et al., 2019) is a comprehensive collection of server telemetry obtained from production clusters within a major Internet corporation. Spanning five consecutive weeks, this multivariate time series dataset captures 38 distinct resource utilization metrics across heterogeneous computing nodes in a distributed cluster environment. Anomalies are annotated by domain experts.

**CT**[1] captures three-dimensional pen tip movement during handwriting tasks. This specialized collection comprises velocity trajectory data for 20 distinct characters, with each character represented by multiple labeled samples. All samples originate from a single writer, ensuring consistent writing style for robust primitive extraction and modeling. Considering the particularity of the handwriting style, letters G, M, Q, W, and Z are regarded as anomaly classes.

---

[1] https://archive.ics.uci.edu/dataset/175/character+trajectories

**SAD**[2] comprises time series of Mel-Frequency Cepstral Coefficients (MFCCs) corresponding to spoken utterances of ten Arabic digits (0-9). This comprehensive collection includes recordings from 88 native Arabic speakers, balanced with 44 male and 44 female participants. Taking into account the speaking rates and styles of spoken digit articulation, digits zero, three, seven, and eight are treated as anomaly classes.

**PTBXL** (Wagner et al., 2020) represents a large-scale, publicly available electrocardiography (ECG) collection encompassing both normal sinus rhythms and pathological cardiac conditions. The dataset is systematically annotated with four major diagnostic categories: Myocardial Infarction (MI), ST/T Change (STTC), Conduction Disturbance (CD), and Hypertrophy (HYP).

**TUSZ** (Shah et al., 2018) represents the largest publicly available electroencephalogram (EEG) dataset specifically curated for seizure detection and classification research. This comprehensive collection encompasses both normal brain activity patterns, primarily captured during resting-state conditions, and abnormal epileptiform activity across diverse seizure types. The dataset is systematically annotated with three major seizure classifications: Focal Non-Specific Seizure (FNSZ), Generalized Non-Specific Seizure (GNSZ), and Complex Partial Seizure (CPSZ), each representing distinct electrophysiological patterns and clinical manifestations.

Unsupervised and open-set settings are both included in our experiments. To verify the robustness, the experiments under both settings were conducted under a contamination level of 2%, where the contaminated samples were extracted from the original test data and explicitly excluded from the final test set, thereby eliminating any potential data leakage. For our method, 20% samples are randomly selected from the training set to form the validation set, and the influence of each training sample is estimated with the validation set using the validation loss.

## F. Implementation Details

In IMPACT, Temporal Convolutional Network (TCN) is utilized as the feature extractor $\phi$. TCN offers distinct advantages over traditional recurrent neural networks, and is widely applied in the field requiring the temporal dependencies. $\phi$ is with one hidden layer, and the two different learning heads $h$ and $h'$ are two-layer multi-layer perceptron networks with ReLU activation. The hidden layer of $\phi$ and learning heads has 64 neural units by default, and the dimension of the feature space is also 64. Our sample selection uses $k$ to control the number of normal reference samples and perturbed samples, and we use $k = 5$ by default. $\alpha$ is to control the perturbation strength, and we set $\alpha = 0.02$ by default. $\lambda$ is set to 1.0 to balance the loss term. For training and retraining process, we conduct the whole stage within ten epochs, where nine epochs are used for training and retraining only requires one epoch.

For all the methods, we set the mini-batch size as 64, the learning rate as 3e-4, and conduct all experiments within 10 epochs. All the methods used in our experiments are implemented in Python. For unsupervised baselines, TCN-AE is sourced from the mvts-ano-eval[3] repository, and THOC is implemented based on a publicly unofficial PyTorch adaptation[4] due to the absence of an official release. We implement TranAD, DCdetector, and COUTA from the DeepOD[5] package. GPT4TS[6] and DADA[7] utilize the authors' original implementations. For open-set competitors, the feature extractor used is exactly the same as that employed in IMPACT. DevNet, DSAD and NCAD are also adopted from DeepOD package. DRA[8], MOSAD[9], InvAD[10], and WSAD-DT [11] are reproduced from their respective official code repositories. All experiments are conducted on a workstation with an Intel(R) Core(TM) i9-10980XE CPU and a single NVIDIA GeForce RTX 3090 GPU.

---

[2]https://archive.ics.uci.edu/dataset/195/spoken+arabic+digit
[3]https://github.com/astha-chem/mvts-ano-eval
[4]https://github.com/carrtesy/THOC-Pytorch
[5]https://github.com/xuhongzuo/DeepOD
[6]https://github.com/DAMO-DI-ML/NeurIPS2023-One-Fits-All
[7]https://github.com/decisionintelligence/DADA
[8]https://github.com/choubo/DRA
[9]https://github.com/Armanfard-Lab/MOSAD
[10]https://github.com/fly-orange/InvAD
[11]https://github.com/Walid10010/Weakly-Supervised-Anomaly-Detection-via-Dual-Tailed-Kernel

# G. Algorithms Details

We outline the detailed procedure of IMPACT in Algorithm 1. IMPACT begins by training the initial model $f$ with the training set $\mathcal{D}$ containing both contaminated normal samples $\mathcal{D}_n$ and labeled anomalies $\mathcal{D}_a$. During the iterative training process, IMPACT identifies contaminated samples through influence function analysis in Step 8, where samples with positive influence values are subjected to label flipping in Step 10. Then several subsets are constructed in Steps 11-24. Among them, $\mathcal{D}_{per}$ contains the top-$k$ normal samples with the largest influence values, and $\mathcal{D}_s$ includes normal samples and identified anomalies (including originally labeled ones and discovered contaminated samples). Most helpful normal samples $\mathcal{D}_h$ are combined with the perturbed features $\mathcal{W}'_{per}$ that embody unseen abnormality. Finally, model is retained via a combined loss function in Step 25.

---

**Algorithm 1** Procedure of IMPACT

---

**Input:** Initial model $f$ with parameters $\theta = \{\theta_\phi, \theta_h\}$, contaminated normal subset $\mathcal{D}_n = \{(\boldsymbol{x}_i, y_i)\}_{i=1}^N$, abnormal subset $\mathcal{D}_a$, training set $\mathcal{D} \doteq \mathcal{D}_n \cup \mathcal{D}_a$, validation set $\mathcal{V}$.
**Output:** Trained model $f'$ with parameters $\hat{\theta}_{re} = \{\hat{\theta}_\phi, \hat{\theta}_h, \hat{\theta}_{h'}\}$
1: Train the model $f$ with $\mathcal{D}$ within $w$ epochs to get optimal parameters $\hat{\theta}$.
2: Initialize $\mathcal{D}_{ref} = \emptyset$
3: **repeat**
4:    Initialize $\mathcal{D}'_{con}, \mathcal{D}_{help}, \mathcal{D}_s, \mathcal{D}_{anom}, \mathcal{W}'_{per} = \emptyset$
5:    Sample mini-batch training data $\mathcal{B} \sim \mathcal{D}$
6:    **for** $\boldsymbol{z}_i \in \mathcal{B}$ **do**
7:        **if** $\boldsymbol{z}_i \in \mathcal{D}_n$ **then**
8:            Calculate the influence of the training sample $\boldsymbol{z}_i$ on $\mathcal{V}$ using Equation (7)
9:            **if** $\mathcal{I}_L(\boldsymbol{z}_i) > 0$ **then**
10:               Label flipping on the identified contaminated training sample with $\boldsymbol{z}_{i\mathbf{1}} \leftarrow \boldsymbol{z}_i$
11:               $\mathcal{D}'_{con} \leftarrow \mathcal{D}'_{con} \cup \{\boldsymbol{z}_{i\mathbf{1}}\}$
12:            **else**
13:               $\mathcal{D}_{help} \leftarrow \mathcal{D}_{help} \cup \{\boldsymbol{z}_i\}, \mathcal{D}_s \leftarrow \mathcal{D}_s \cup \{\boldsymbol{z}_i\}$
14:            **end if**
15:        **else**
16:            $\mathcal{D}_{anom} \leftarrow \mathcal{D}_{anom} \cup \{\boldsymbol{z}_i\}, \mathcal{D}_s \leftarrow \mathcal{D}_s \cup \{\boldsymbol{z}_i\}$
17:        **end if**
18:    **end for**
19:    $\mathcal{D}_{per} \leftarrow \mathcal{D}_{help}|_{\text{top-k largest}}, \mathcal{D}_{ref} \leftarrow \mathcal{D}_{ref} \cup \mathcal{D}_{help}|_{\text{top-k smallest}}, \mathcal{D}_{clean} \leftarrow \mathcal{D}_{help} \cup \mathcal{D}_{anom}$
20:    $\mathcal{D}_s \leftarrow \mathcal{D}_s \cup \mathcal{D}'_{con}, \mathcal{D}_h \leftarrow \mathcal{D}_{help} \setminus \mathcal{D}_{per}$
21:    **for** $(\boldsymbol{x}_i, 0) \in \mathcal{D}_{per}$ **do**
22:        Perturbing the embedded feature $\boldsymbol{\varphi}_i$ of $\boldsymbol{x}_i$ to obtain $\boldsymbol{\varphi}_{i\zeta_i}$ using Equation (15)
23:        $\mathcal{W}'_{per} \leftarrow \mathcal{W}'_{per} \cup \{(\boldsymbol{\varphi}_{i\zeta_i}, 1)\}$
24:    **end for**
25:    Update network parameters according to Equation (19) with $\mathcal{D}_s$ and $\mathcal{D}_h \cup \mathcal{W}'_{per}$
26: **until** Reach maximum number of mini-batches
27: **return:** $f'$ with parameters $\hat{\theta}_{re}$

---

# H. Additional Empirical Results

## H.1. Detection Performance of Seen Versus Unseen Anomalies

Although the results reported in Table 2 have highlighted the superiority of IMPACT for detecting seen and unseen anomalies together, it is still necessary to further explore the performance of IMPACT and several advanced baselines on the seen and unseen anomaly classes separately. As shown in Table 5, IMPACT demonstrates strong competitiveness on both seen and unseen anomaly detection across all datasets. While WSAD-DT excel on seen anomalies in CT and TUSZ but struggles with unseen ones, IMPACT achieves balanced performance on both tasks, confirming its capability to simultaneously optimize for seen anomaly discrimination and unseen anomaly generalization.

*Table 5.* Detection accuracy (AUC ± standard deviation) of IMPACT and its competing methods w.r.t sample types (seen and unseen anomalies). All results are in %. The best ones are **boldfaced** and the second best are underlined.

| Dataset | seen anomaly | Seen | | | | Unseen | | | |
|---|---|---|---|---|---|---|---|---|---|
| | | MOSAD | InvAD | WSAD-DT | Ours | MOSAD | InvAD | WSAD-DT | Ours |
| CT | Letter-G | $53.67_{\pm5.80}$ | $51.94_{\pm2.74}$ | $97.12_{\pm0.31}$ | **$99.60_{\pm0.22}$** | $74.53_{\pm0.38}$ | $72.18_{\pm0.75}$ | $60.70_{\pm2.32}$ | **$75.53_{\pm3.90}$** |
| | Letter-M | $94.47_{\pm0.78}$ | $85.33_{\pm3.75}$ | $98.39_{\pm0.31}$ | **$99.82_{\pm0.01}$** | $71.26_{\pm0.84}$ | $73.88_{\pm1.07}$ | $67.93_{\pm2.78}$ | **$75.04_{\pm5.16}$** |
| | Letter-Q | $75.06_{\pm2.25}$ | $71.78_{\pm2.87}$ | $96.00_{\pm1.02}$ | **$99.89_{\pm0.04}$** | $73.11_{\pm1.11}$ | **$75.73_{\pm1.25}$** | $69.12_{\pm2.69}$ | $73.27_{\pm4.11}$ |
| | Letter-W | $74.21_{\pm2.13}$ | $68.23_{\pm2.30}$ | $81.65_{\pm2.89}$ | **$95.52_{\pm0.53}$** | $78.50_{\pm0.31}$ | $77.33_{\pm0.85}$ | $65.73_{\pm1.86}$ | **$79.16_{\pm2.95}$** |
| | Letter-Z | $97.22_{\pm1.57}$ | $91.43_{\pm1.75}$ | $99.91_{\pm0.01}$ | **$99.99_{\pm0.01}$** | $71.87_{\pm0.95}$ | $69.39_{\pm1.00}$ | $65.73_{\pm1.86}$ | **$74.34_{\pm2.70}$** |
| | Mean | $78.93_{\pm15.82}$ | $73.74_{\pm13.83}$ | $94.61_{\pm6.61}$ | **$98.96_{\pm1.73}$** | $73.85_{\pm2.58}$ | $73.70_{\pm2.76}$ | $67.71_{\pm4.67}$ | **$75.47_{\pm2.00}$** |
| SAD | Digit-Zero | $26.78_{\pm0.90}$ | $49.55_{\pm4.21}$ | **$83.86_{\pm1.32}$** | $77.08_{\pm6.55}$ | **$61.68_{\pm1.02}$** | $51.03_{\pm3.41}$ | $52.76_{\pm2.42}$ | $58.11_{\pm5.23}$ |
| | Digit-Three | $62.84_{\pm0.71}$ | $53.89_{\pm2.95}$ | $73.87_{\pm3.25}$ | **$79.25_{\pm4.70}$** | $49.65_{\pm0.52}$ | $48.49_{\pm1.62}$ | **$69.05_{\pm0.48}$** | $63.36_{\pm4.83}$ |
| | Digit-Seven | $39.36_{\pm1.69}$ | $44.89_{\pm7.52}$ | $61.82_{\pm1.46}$ | **$63.68_{\pm3.70}$** | $57.13_{\pm0.99}$ | $51.61_{\pm2.35}$ | $63.99_{\pm1.30}$ | **$73.05_{\pm2.11}$** |
| | Digit-Eight | $82.88_{\pm1.10}$ | $50.89_{\pm4.17}$ | $74.47_{\pm1.27}$ | **$97.96_{\pm0.58}$** | $43.26_{\pm0.83}$ | $51.62_{\pm1.37}$ | $51.66_{\pm2.44}$ | **$52.50_{\pm4.13}$** |
| | Mean | $52.97_{\pm21.58}$ | $49.81_{\pm3.24}$ | $78.51_{\pm12.08}$ | **$79.49_{\pm12.22}$** | $52.93_{\pm7.04}$ | $50.69_{\pm1.29}$ | $59.37_{\pm7.39}$ | **$61.76_{\pm7.57}$** |
| PTBXL | MI | $66.19_{\pm4.98}$ | $52.34_{\pm5.65}$ | $67.97_{\pm1.55}$ | **$69.37_{\pm5.29}$** | $63.91_{\pm4.99}$ | $53.76_{\pm4.56}$ | $62.94_{\pm1.52}$ | **$64.90_{\pm2.94}$** |
| | STTC | $57.42_{\pm4.24}$ | $53.51_{\pm4.89}$ | $56.05_{\pm2.10}$ | **$59.85_{\pm3.64}$** | $56.12_{\pm4.83}$ | $54.74_{\pm5.53}$ | $56.48_{\pm3.14}$ | **$59.59_{\pm4.11}$** |
| | CD | $62.89_{\pm4.06}$ | $51.84_{\pm7.04}$ | **$67.38_{\pm1.26}$** | $64.37_{\pm7.14}$ | $62.85_{\pm3.62}$ | $51.34_{\pm7.28}$ | **$64.86_{\pm0.75}$** | $61.92_{\pm5.25}$ |
| | HYP | $64.64_{\pm3.50}$ | $54.16_{\pm7.71}$ | $67.59_{\pm2.87}$ | **$68.07_{\pm3.04}$** | $61.73_{\pm2.79}$ | $52.03_{\pm6.23}$ | $58.63_{\pm2.38}$ | **$64.91_{\pm3.62}$** |
| | Mean | $62.79_{\pm3.31}$ | $52.96_{\pm0.92}$ | $64.75_{\pm5.03}$ | **$65.42_{\pm3.70}$** | $61.15_{\pm3.01}$ | $52.97_{\pm1.35}$ | $60.73_{\pm3.33}$ | **$62.83_{\pm2.23}$** |
| TUSZ | FNSZ | $73.92_{\pm1.48}$ | $57.14_{\pm7.39}$ | $73.02_{\pm0.89}$ | **$79.54_{\pm1.40}$** | $64.46_{\pm4.57}$ | $57.17_{\pm10.99}$ | $53.76_{\pm3.60}$ | **$71.90_{\pm5.44}$** |
| | GNSZ | $62.91_{\pm3.35}$ | $53.48_{\pm11.11}$ | $84.49_{\pm0.75}$ | **$85.95_{\pm2.23}$** | $68.33_{\pm1.74}$ | $55.67_{\pm7.67}$ | $47.01_{\pm2.20}$ | **$69.42_{\pm2.77}$** |
| | CPSZ | $60.45_{\pm5.08}$ | $58.51_{\pm10.26}$ | $84.66_{\pm1.92}$ | **$89.57_{\pm3.12}$** | $70.95_{\pm2.16}$ | $55.71_{\pm9.04}$ | $52.89_{\pm1.72}$ | **$74.52_{\pm3.61}$** |
| | Mean | $65.76_{\pm5.86}$ | $56.38_{\pm2.12}$ | $80.72_{\pm5.45}$ | **$85.02_{\pm4.15}$** | $67.91_{\pm2.67}$ | $56.18_{\pm0.70}$ | $51.22_{\pm3.00}$ | **$71.95_{\pm2.08}$** |

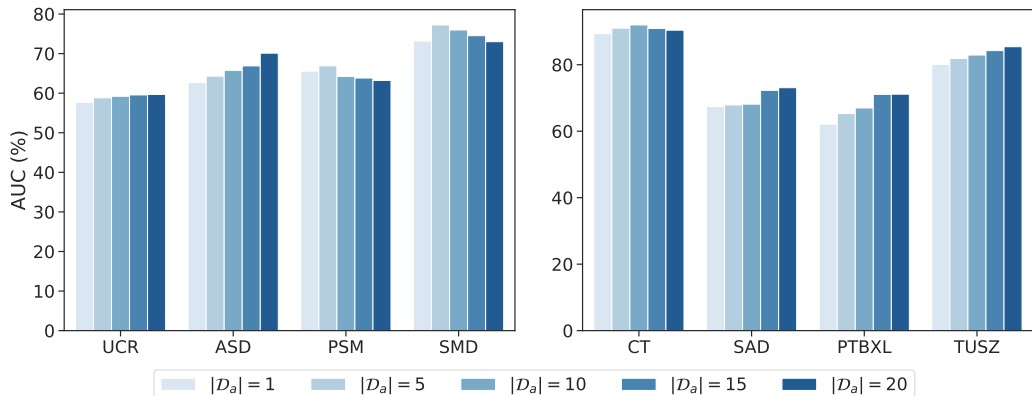

*Figure 4.* AUC performance of IMPACT w.r.t different number of labeled anomalies under the general setting.

## H.2. Utility of Few-Shot Labeled Anomalies

This experiment investigates how effective our method is in utilizing the provided few-shot anomaly samples. We choose the number of labeled anomalies $|\mathcal{D}_a|$ from $\{1, 5, 10, 15, 20\}$ under both general and hard settings, and the results are presented in Figure 4 and Figure 5 respectively. For general setting, the performance on most datasets improves as the number of labeled anomalies increases, demonstrating that our method effectively leverages the provided anomalies. However, datasets such as PSM and SMD exhibit performance degradation with additional anomaly samples, likely due to their relatively simple anomaly patterns, which induce model overfitting. This phenomenon similarly manifests in the hard setting, increasing available anomalies in CT and SAD impedes generalization to unseen anomalies, which suggests that excessive supervision on single anomaly class constrains the model's capacity to capture diverse unknown patterns. Conversely, for larger and more complex datasets like PTBXL and TUSZ, additional anomaly samples consistently enhance detection performance, indicating that richer supervised anomaly knowledge in complex scenarios facilitates more robust and generalizable anomaly detection.

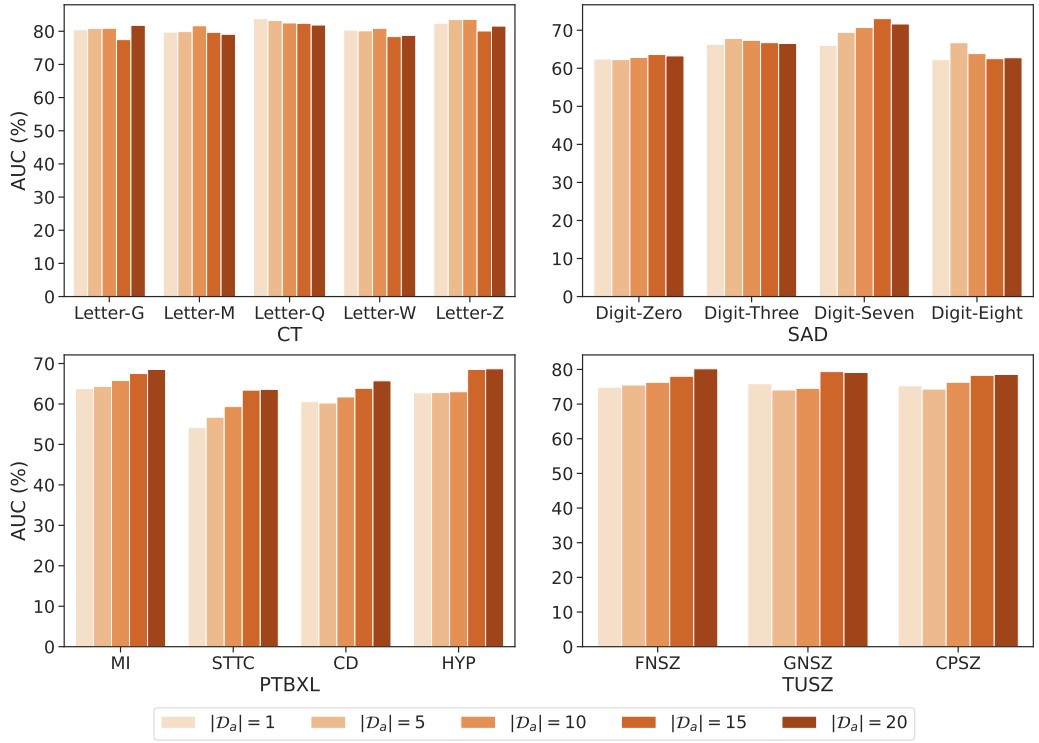

*Figure 5.* AUC performance of IMPACT w.r.t different number of labeled anomalies under the hard setting.

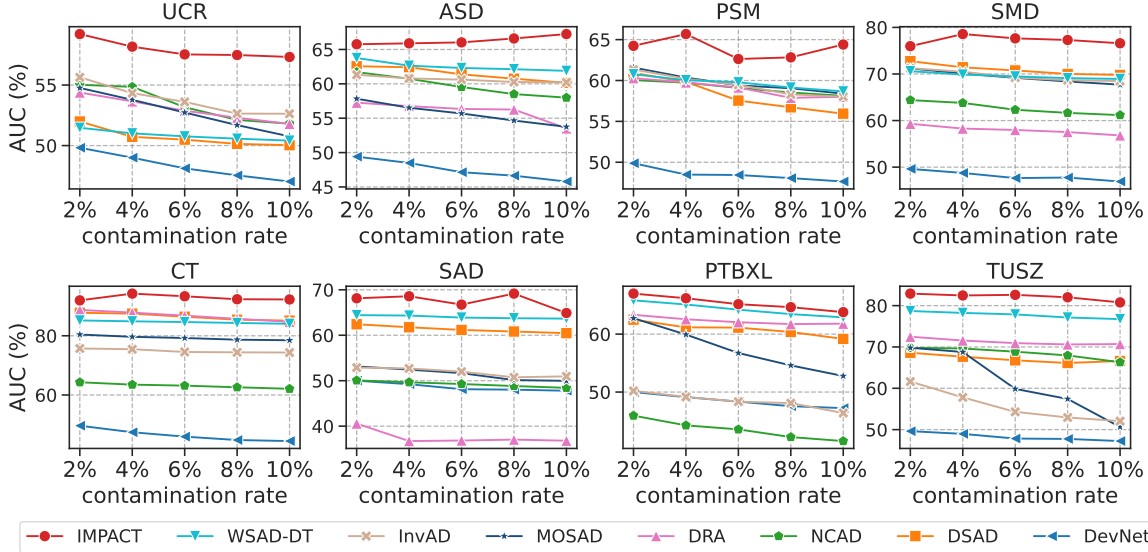

*Figure 6.* AUC performance of IMPACT and its competing baselines w.r.t different contamination rates under the general setting.

## H.3. Effectiveness of Handling Anomaly Contamination

Existing open-set and supervised anomaly detection methods mainly focus on how to leverage provided abnormal knowledge, while simply treating the remaining unlabeled training data as normal data. However, in real-world scenarios, training data may be inadvertently contaminated by potential anomalies, which mislead the model during learning. Therefore, we examine the effectiveness of handling anomaly contamination under different contamination rates. A small portion of the test anomalies is selected to contaminate the training set, with the contamination rate ranging from 2% to 10%.

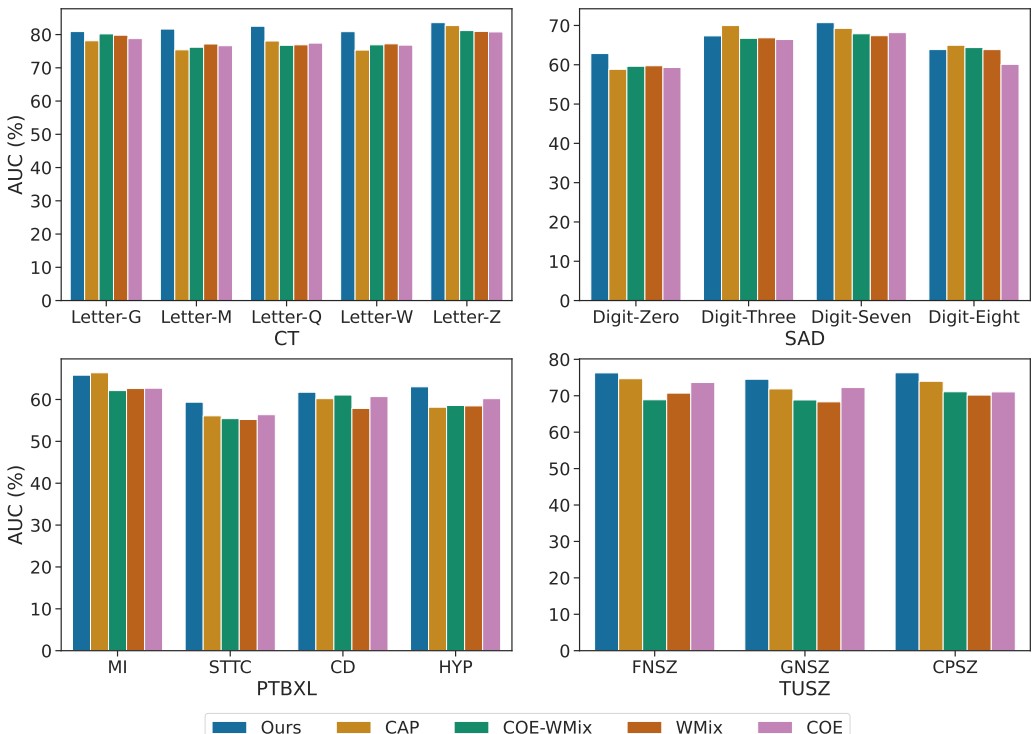

*Figure 7.* AUC results w.r.t. various methods to create pseudo anomalies under the hard setting.

Figure 6 depicts the AUC performance of eight anomaly detection methods under general setting. IMPACT demonstrates consistent advantages in various scenarios. The other competing baselines essentially treat the unlabeled data as the normal class in a binary classification task, thereby identifying anomalies through classification during the testing phase. However, when the training data is contaminated, the model learns biased patterns that incorrectly identify test anomalies resembling the contaminating samples as normal, leading to elevated false negative rates. As anticipated, the detection performance of these methods deteriorates progressively with increasing contamination rates. In contrast, our approach employs the influence function to identify potential contaminated samples and relabels them as true anomalies, thereby converting detrimental contamination into beneficial anomaly knowledge. Experimental results confirm that our method maintains substantially more stable performance across all datasets. Notably, on certain datasets such as ASD and SMD, performance even improves as contamination rate increases, indicating that the repurposing of contaminated samples into supervised anomaly examples effectively enhances the model's capacity for abnormality learning.

### H.4. Effect of Anomaly Generation Methods

Current methods usually generate pseudo anomalies during training to simulate possible classes of unseen anomalies, aiming to generalize to unknown anomalies in the test phase. To validate that our risk-reduction-based anomaly generation more effectively facilitates generalization to unseen anomalies, we further evaluate four alternative anomaly augmentation techniques, including Contextual Outlier Exposure (COE) (Carmona et al., 2022), Window Mixup (WMix) (Carmona et al., 2022), COE-WMix that utilizes both COE and WMix, and CutAddPaste (CAP) (Wang et al., 2024).

Under the hard setting, we substitute each of the above-mentioned techniques into our framework while keeping other components unchanged, and the results are shown in Figure 7. It is obvious that our methods achieve more stable and much better performance than other anomaly augmentation methods. This advantage stems from a critical limitation of existing augmentation methods: they cannot guarantee the quality of generated pseudo anomalies. Low-quality pseudo anomalies exhibit limited coverage of real unseen anomaly distributions and may provide misleading information far from actual anomalies, potentially causing models to overfit to incorrect abnormal knowledge. In contrast, our feature perturbation approach directly modifies embedded features, theoretically ensuring that generated samples diverge meaningfully from known patterns while remaining within plausible anomaly manifolds.

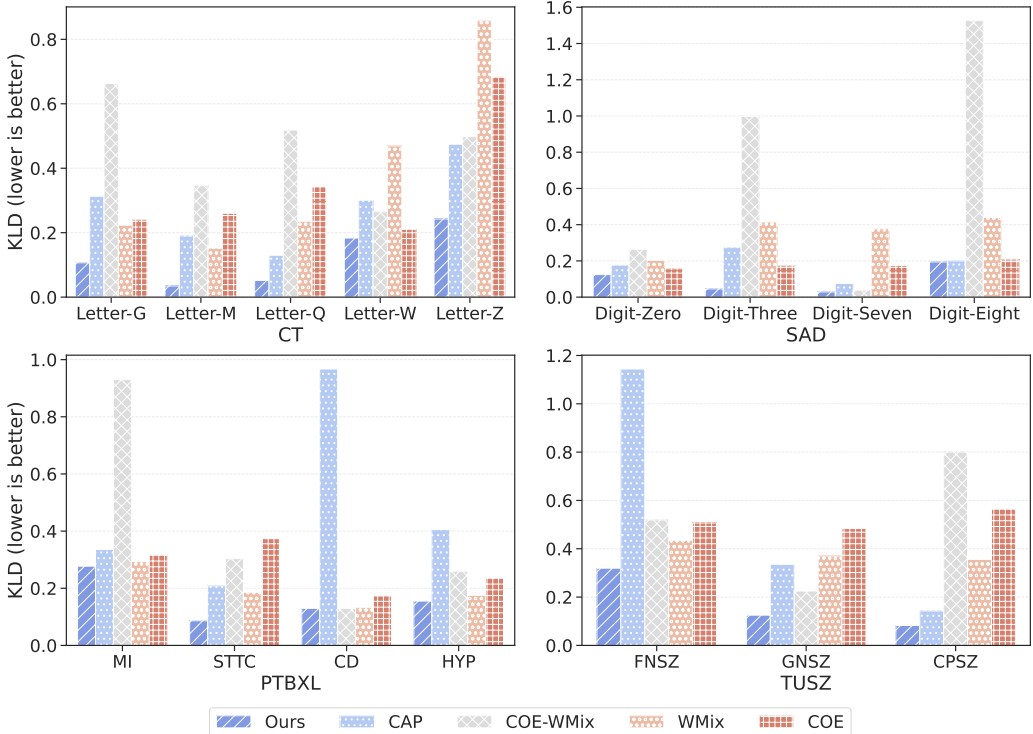

*Figure 8.* Kullback-Leibler Divergence (KLD) of anomaly features from various datasets under the hard setting. $KLD(\Phi_{unseen}||\Phi_{pseudo})$ is given, which implies distribution discrepancies of unseen anomaly features and the generated pseudo anomaly features.

We further quantify the quality of generated anomalies through distributional analysis. As illustrated in Figure 8, our method achieves the lowest Kullback-Leibler Divergence (KLD) values across all datasets, indicating that our generated pseudo anomalies exhibit the closest distributional alignment with real unseen anomalies. Specifically, alternative methods (CAP, COE-WMix, WMix, and COE) produce most KLD values ranging from 0.15 to 1.53 across different datasets, and our approach maintains consistently lower values between 0.03 and 0.32, representing an average reduction of 64.13% in distributional discrepancy. This distributional superiority directly corresponds to previously demonstrated improvement in model performance. The lower KLD values correspond to better coverage of the unseen anomaly manifold, ensuring that models trained with our generated anomalies develop more accurate decision boundaries. Traditional augmentation methods, particularly those relying on simple feature mixing or interpolation (*e.g.*, WMix and COE-WMix), often generate anomalies that cluster in limited regions of the feature space, failing to capture the full diversity of real anomalous patterns. Consequently, models trained on such limited distributions exhibit reduced generalization capabilities when encountering unseen anomalies.

Finally, we analyze the differences among the seen, unseen, and generated pseudo anomalies, and depict the feature t-SNE visualization in Figure 9, where the features are extracted under the condition that Letter-G serves as the seen anomaly class in the CT dataset. It can be observed that the pseudo anomalies generated by COE predominantly cluster near seen anomalies, with some scattered around normal samples but showing minimal overlap with unseen anomalies. Conversely, the pseudo anomalies produced by our method exhibit substantial divergence from seen anomalies while aligning closely with the actual unseen anomaly distribution. This visualization confirms that our feature perturbation strategy generates pseudo anomalies that better approximate the real unseen anomalies, thereby facilitating more effective generalization in challenging open-set scenarios.

### H.5. Visualization of Identified Helpful and Harmful Samples

We provide examples of helpful and harmful samples identified by influence functions to illustrate the effectiveness of establishing the referable normality and eliminating potential contamination. As shown in Figure 10, the visualization results reveal that the most helpful samples are typically normal samples exhibiting morphological divergence from the given test

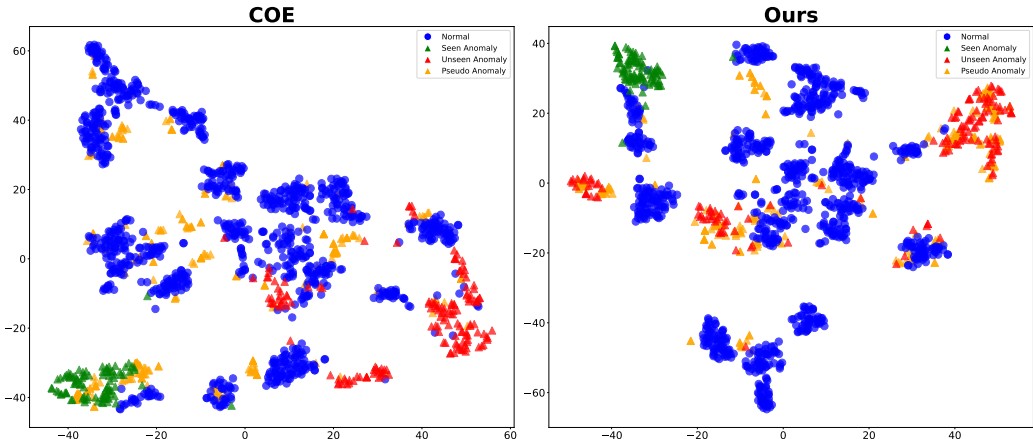

*Figure 9.* Feature t-SNE visualization. The features are extracted under the condition that Letter-G serves as the seen anomaly class in the CT dataset.

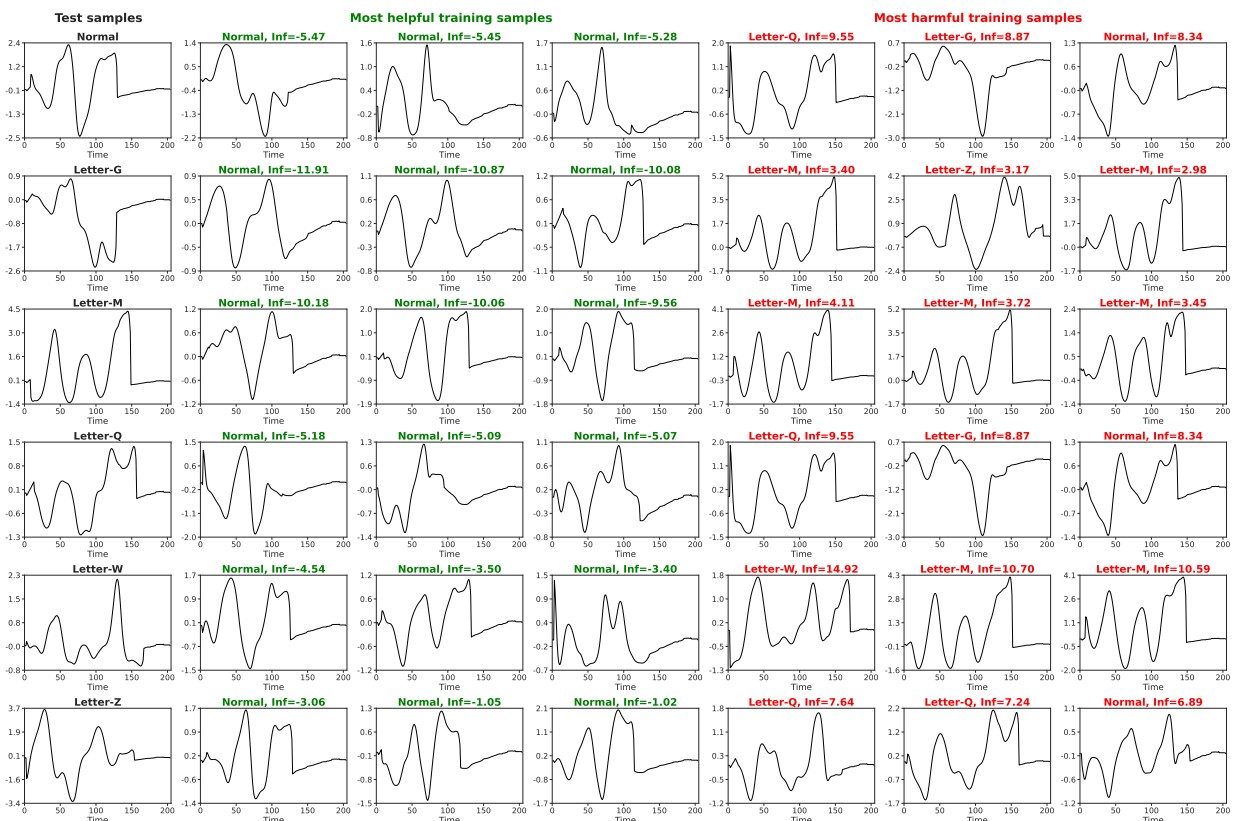

*Figure 10.* Identified helpful and harmful training samples from CT. For the test samples from normal class and five distinct anomaly classes, each corresponds to three helpful training samples with the minimum influence estimates and three harmful training samples with the highest influence estimates.

sample. The underlying rationale is that such diverse normal samples compel the model to learn from a broader spectrum of feature variations, thereby enhancing its discriminative capacity across heterogeneous normal patterns. Conversely, the most harmful samples are those morphologically similar to the test sample yet inherently abnormal (*i.e.*, contaminated samples). These samples introduce conflicting supervisory signals that misguide the optimization trajectory, causing the model to converge toward biased decision boundaries and consequently degrading detection performance.

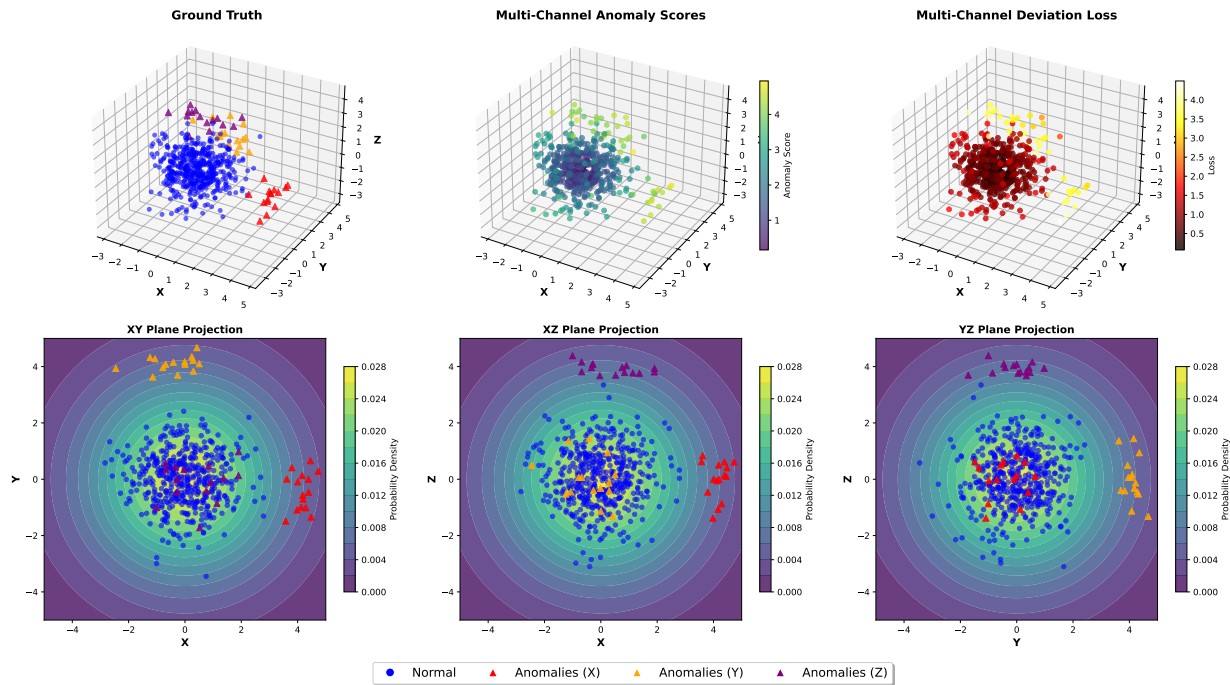

*Figure 11.* A qualitative example from the PTBXL dataset to demonstrate how multi-channel strategy helps to identify channel-specific anomalies that may be missed in single-channel analysis. The first row displays three-dimensional visualizations of ground truth data distribution, multi-channel anomaly scores, and multi-channel deviation losses. The second row presents different plane projections, which indicate that some anomalies are difficult to distinguish from a single projection, thereby highlighting the importance of multi-channel anomaly discrimination.

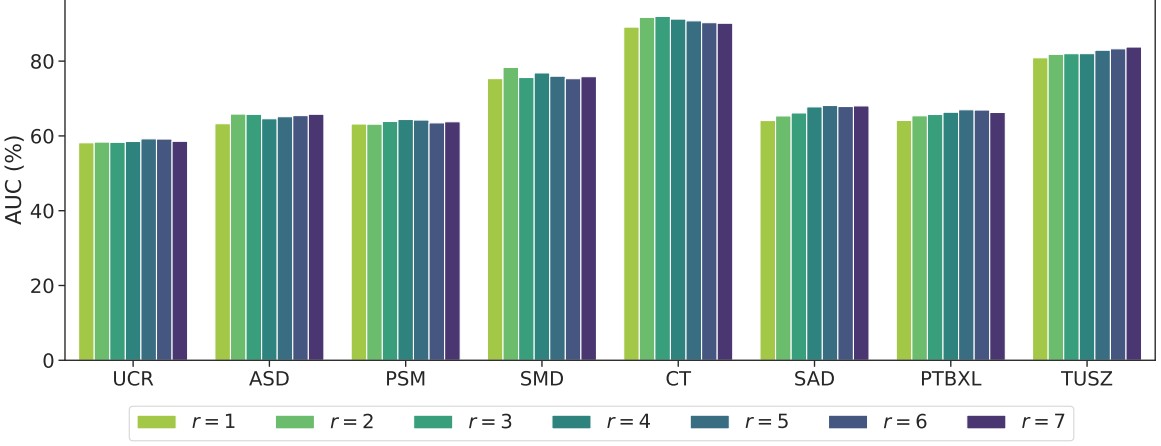

*Figure 12.* AUC performance of IMPACT w.r.t different number of channels under the general setting.

## H.6. Interpretation of Multi-Channel Strategy

Recall Theorem 1, we extend the deviation loss (Pang et al., 2019) by proposing a multi-channel strategy and provide a theoretical explanation from the perspective of entropy minimization. In this experiment, we further empirically inspect how each individual channel contributes to the anomaly detection process. As shown in Figure 11, the qualitative results demonstrate that different channels capture distinct aspects of anomalies and some anomalies are strongly manifested in specific channels while barely detectable in others. For instance, anomalies primarily visible in the X-Y plane projection are nearly indistinguishable in the X-Z view, illustrating the complementary nature of multi-channel information. Our multi-channel strategy naturally assesses the abnormality degree of samples from multiple perspectives, thereby maintaining

*Table 6.* Point-level anomaly detection results (Affiliation-F1/VUS-ROC $\pm$ standard deviation) of IMPACT and its competing methods under the unsupervised and open-set general settings. All results are in %. The best ones are **boldfaced** and the second best are underlined.

| | Methods | Affiliation-F1 | | | | VUS-ROC | | | |
|---|---|---|---|---|---|---|---|---|---|
| | | UCR | ASD | PSM | SMD | UCR | ASD | PSM | SMD |
| Unsupervised | TCN-AE | $67.30_{\pm0.54}$ | $70.50_{\pm0.05}$ | $73.47_{\pm0.73}$ | $64.76_{\pm0.19}$ | $51.89_{\pm0.68}$ | $62.62_{\pm0.17}$ | $60.44_{\pm0.77}$ | $64.17_{\pm0.06}$ |
| | THOC | $67.34_{\pm0.11}$ | $70.84_{\pm0.76}$ | $72.50_{\pm1.53}$ | $74.85_{\pm1.00}$ | $53.48_{\pm0.51}$ | $61.65_{\pm1.46}$ | $55.35_{\pm2.56}$ | $64.42_{\pm0.77}$ |
| | TranAD | $66.93_{\pm0.18}$ | $71.40_{\pm0.45}$ | $76.32_{\pm0.44}$ | $68.62_{\pm0.00}$ | $50.20_{\pm0.56}$ | $62.54_{\pm0.57}$ | $63.72_{\pm0.35}$ | $68.25_{\pm0.17}$ |
| | DCdetector | $66.67_{\pm0.00}$ | $69.78_{\pm0.00}$ | $70.77_{\pm0.09}$ | $68.43_{\pm0.11}$ | $52.04_{\pm1.86}$ | $54.85_{\pm0.17}$ | $52.21_{\pm0.14}$ | $54.89_{\pm0.47}$ |
| | GPT4TS | $67.92_{\pm0.30}$ | $71.50_{\pm0.63}$ | $72.78_{\pm0.41}$ | $70.43_{\pm0.21}$ | $61.09_{\pm0.36}$ | $66.20_{\pm0.41}$ | $59.80_{\pm0.46}$ | $72.71_{\pm0.30}$ |
| | COUTA | $66.99_{\pm0.46}$ | $70.41_{\pm1.25}$ | $78.71_{\pm0.23}$ | $66.34_{\pm0.03}$ | $55.55_{\pm1.51}$ | $65.96_{\pm1.14}$ | $67.02_{\pm4.32}$ | $65.67_{\pm1.60}$ |
| | DADA | $66.67_{\pm0.00}$ | $69.79_{\pm0.00}$ | **$86.93_{\pm3.88}$** | $67.82_{\pm0.00}$ | $54.69_{\pm0.03}$ | $60.96_{\pm0.03}$ | $73.12_{\pm0.05}$ | $67.60_{\pm0.00}$ |
| Open-Set | DevNet | $66.55_{\pm0.00}$ | $70.38_{\pm0.21}$ | $33.09_{\pm2.55}$ | $68.45_{\pm1.52}$ | $48.72_{\pm2.58}$ | $52.01_{\pm2.89}$ | $48.64_{\pm2.35}$ | $50.58_{\pm0.94}$ |
| | DSAD | $67.04_{\pm0.24}$ | $76.05_{\pm0.46}$ | $76.59_{\pm0.36}$ | $72.73_{\pm2.52}$ | $51.20_{\pm0.78}$ | $76.01_{\pm0.36}$ | $63.47_{\pm1.67}$ | $79.79_{\pm1.19}$ |
| | NCAD | $68.85_{\pm0.16}$ | $74.96_{\pm0.66}$ | $79.56_{\pm1.32}$ | $80.03_{\pm0.75}$ | $60.01_{\pm0.35}$ | $70.64_{\pm1.04}$ | $61.41_{\pm1.42}$ | $68.80_{\pm0.61}$ |
| | DRA | $67.35_{\pm0.36}$ | $71.13_{\pm0.86}$ | $73.09_{\pm1.44}$ | $70.14_{\pm1.36}$ | $59.64_{\pm0.55}$ | $58.84_{\pm0.07}$ | $52.40_{\pm0.33}$ | $55.90_{\pm0.77}$ |
| | MOSAD | $67.24_{\pm0.02}$ | $71.77_{\pm0.07}$ | $73.17_{\pm0.12}$ | $69.26_{\pm0.07}$ | $55.28_{\pm0.02}$ | $62.28_{\pm0.08}$ | $57.45_{\pm0.20}$ | $68.93_{\pm0.54}$ |
| | InvAD | $67.06_{\pm0.02}$ | $70.94_{\pm0.25}$ | $72.97_{\pm0.64}$ | $71.79_{\pm0.32}$ | $58.18_{\pm0.12}$ | $61.25_{\pm1.49}$ | $54.07_{\pm0.82}$ | $62.00_{\pm1.58}$ |
| | WSAD-DT | $68.60_{\pm0.53}$ | $73.57_{\pm0.98}$ | $72.58_{\pm0.37}$ | $71.51_{\pm0.74}$ | $50.04_{\pm0.94}$ | $68.74_{\pm1.12}$ | $53.79_{\pm1.01}$ | $69.57_{\pm0.93}$ |
| | IMPACT | **$72.02_{\pm0.24}$** | **$77.12_{\pm0.18}$** | $86.52_{\pm0.63}$ | **$81.22_{\pm0.21}$** | **$64.53_{\pm0.01}$** | **$78.35_{\pm0.94}$** | **$76.52_{\pm0.84}$** | **$81.37_{\pm0.29}$** |

sensitivity to various types of anomalies. Meanwhile, we also conduct a further study on the impact of the number of channels on the effectiveness of anomaly detection. As illustrated in Figure 12, multi-channel configurations yield substantial performance gains over single-channel setting. However, employing an excessive number of channels may increase model sensitivity to noise, leading to performance degradation. Based on our experimental observations, a configuration of 3–5 channels provides an optimal balance between information richness and noise robustness for most applications.

### H.7. Point-Level Anomaly Detection

Open-set time series anomaly detection treats each time interval as an individual sample and provides a few time interval-level anomaly labels, since acquiring point-level labels is laborious and prone to inaccuracies in practice. When point-level anomaly detection is required, we adopt the standard practice (Xu et al., 2024; Shentu et al., 2025) of employing a sliding window with a step size of 1 to partition the test sequence into consecutive intervals. The anomaly score computed for each interval is then assigned to its final timestep, thereby generating pointwise anomaly predictions while maintaining temporal continuity. We adopt the widely used Affiliation-F1 score (Huet et al., 2022) and VUS-ROC (Boniol et al., 2025) to evaluate the point-level detection performance. As shown in Table 6, IMPACT consistently outperforms both unsupervised and open-set baselines across multiple datasets. Specifically, IMPACT achieves the best Affiliation-F1 scores on three out of four datasets (UCR, ASD, and SMD) and the second best on PSM. In terms of VUS-ROC, IMPACT obtains the highest scores on all four datasets. Notably, on the UCR dataset, IMPACT improves the VUS-ROC by a significant margin (from 61.09% by GPT4TS to 64.53% by ours). The consistent performance gains across different datasets and evaluation metrics validate the proposed framework's ability to achieve more accurate and robust point-level anomaly detection. In addition, Figure 13 and Figure 14 further illustrate the point-level anomaly detection results on univariate and multivariate time series data, respectively. Our method demonstrates superior capability in generating more precise pointwise anomaly scores for both univariate and multivariate data within open-set environments.

### H.8. Complexity Analysis

According to (Koh & Liang, 2017), the time complexity of calculating influence function on all training samples is $O(MP)$, where $M$ is the size of normal subset and $P$ stands for the size of the spatial set formed by the model parameters $\theta$. Note that the time complexity of selecting, flipping, and perturbing one sample is $O(1)$. We can effectively conduct our method in $O(MP)$ time, and further measure the execution time to empirically validate the computational efficiency of IMPACT.

We conduct scalability experiments using two groups of synthetic datasets. The first group maintains a fixed sample size of 2,000 while varying dimensionality across $\{8, 16, 32, 64, 128, 256, 512\}$. The second group holds dimensionality constant

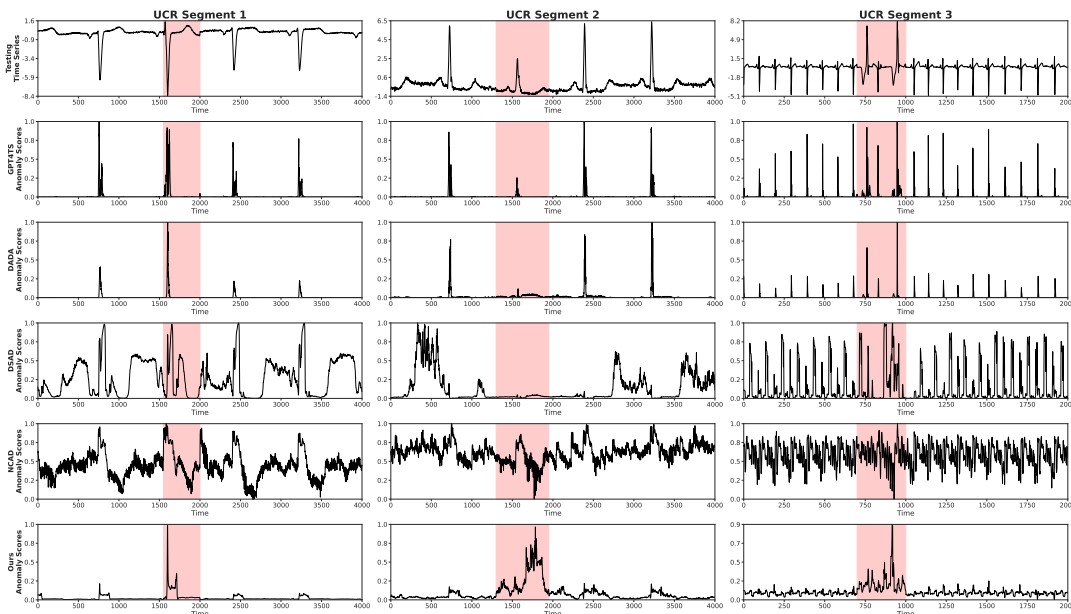

*Figure 13.* Visualization of point-level anomaly scores in comparison with SOTA baselines on the UCR dataset. The first row displays the testing time series, while the remaining rows show the point-level anomaly scores generated by each method. Red regions highlight the ground truth anomaly intervals.

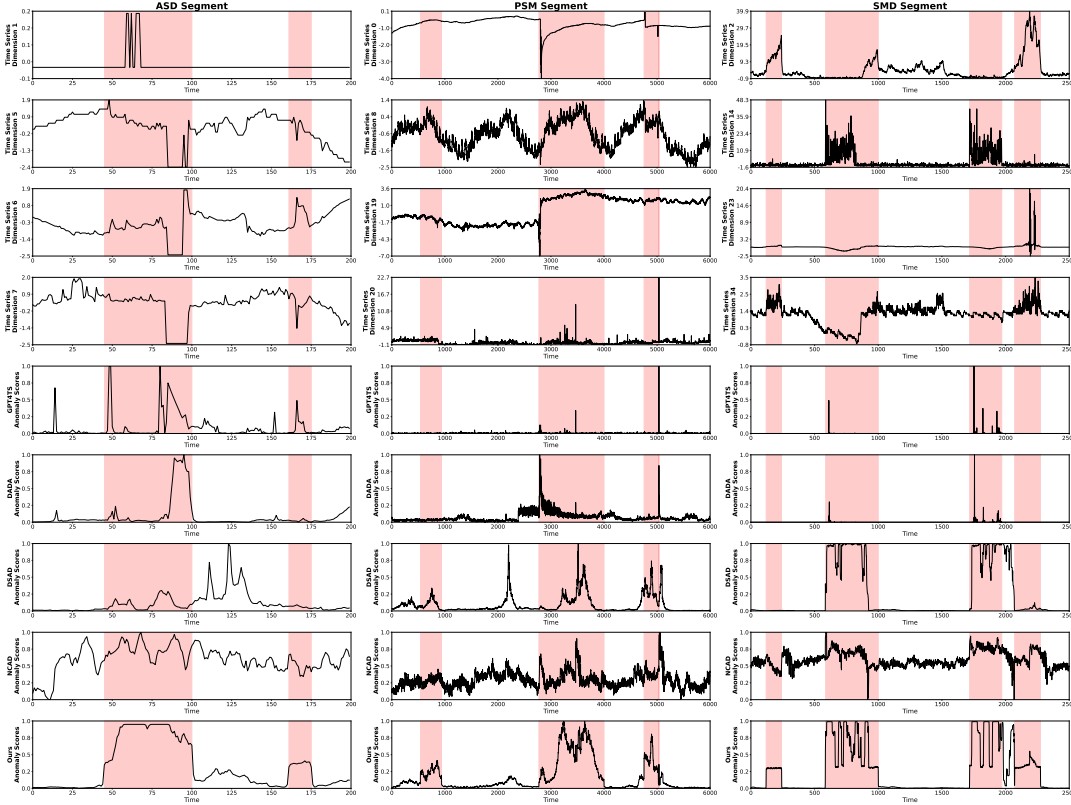

*Figure 14.* Visualization of point-level anomaly scores in comparison with SOTA baselines on the multivariate datasets with several selected dimensions. The first four rows display the selected dimensions of the testing time series, which represent the major anomaly patterns in the data. The last three rows show the point-level anomaly scores generated by each method. Red regions highlight the ground truth anomaly intervals.

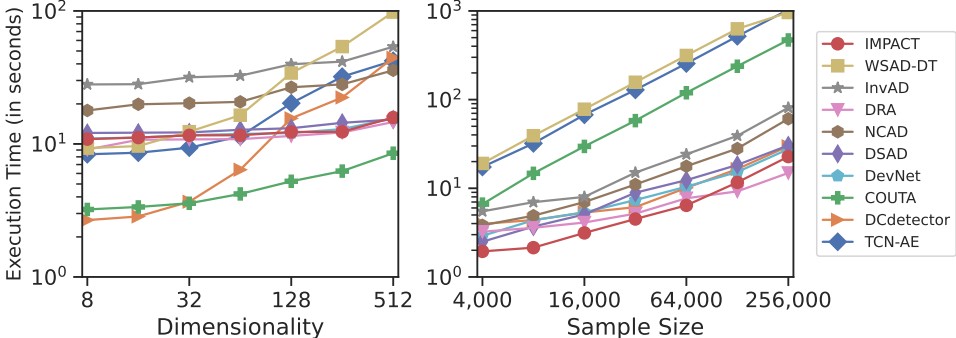

*Figure 15.* Computational efficiency results across different dimensionality and sample sizes.

at 8 while exponentially increasing sequence length from 4,000 to 256,000. For fair comparison, all deep learning baselines are implemented consistently within the PyTorch framework. As shown in Figure 15, IMPACT demonstrates superior computational efficiency compared to other deep learning baselines across both experimental settings. When dimensionality increases to 512 with fixed sample size, IMPACT exhibits only modest execution time growth, confirming its effectiveness in high-dimensional scenarios. With exponentially increasing sample sizes up to 256,000 at fixed dimensionality, IMPACT maintains approximately linear time complexity.

## I. Future Directions

We propose a novel influence-based framework IMPACT which demonstrates significant improvements in robustness and generalization for open-set time series anomaly detection. This section notes four future directions that are worth exploring: (i) IMPACT can be extended to handle multi-modal data by designing appropriate strategies for influence modeling and feature perturbation, as anomalies may manifest across logs, images and videos in industrial and healthcare applications. (ii) IMPACT can be generalized to different learning tasks like pre-training representation models on large-scale time series data. (iii) IMPACT can be adopted to an online learning setting, where the influence of individual samples is updated incrementally to handle concept drift without full retraining. This would enable continuous adaptation to evolving normal and abnormal patterns in real-world dynamic systems. (iv) IMPACT may misidentify rare normal subpopulations that have no semantic overlap with other normal samples as anomalies, representing a group-robustness failure mode. Incorporating explicit subpopulation discovery or group-aware validation set construction prior to influence estimation could further strengthen group robustness.

