# OpenReview forum: "IMPACT: Influence Modeling for Open-Set Time Series Anomaly Detection"
_ICML.cc/2026/Conference — ICML 2026 regular_

### Official Review · Reviewer_QuHg · 2026-03-04

**Soundness:** 2
**Presentation:** 3
**Significance:** 3
**Originality:** 2
**Overall Recommendation:** 4
**Confidence:** 3

**Summary:**

This paper proposes IMPACT, a framework for open-set time series anomaly detection that uses influence functions to (i) identify and relabel contaminated training samples via label flipping and (ii) generate pseudo unseen anomalies through feature perturbation of boundary normal samples. The method is evaluated on eight real-world time series datasets across unsupervised and open-set settings, reporting improvements over several baselines.

**Compliance With Llm Reviewing Policy:**

Affirmed.

**Final Justification:**

The paper proposes IMPACT, a framework that uses influence functions for simultaneous decontamination and pseudo-anomaly generation in open-set time series anomaly detection. The core design — using influence-guided label flipping and feature perturbation from a unified risk-reduction perspective — is well-motivated and, after the rebuttal, well-supported.

**Strengths.** The dual-purpose use of influence functions is a novel system design contribution. The experimental evaluation is extensive (8 datasets, 14 baselines, ablations, scalability analysis). Theorems 2 and 4 provide principled justification for the two key operations.

**Weaknesses and rebuttal assessment.** My main concerns were: (W1) overselling the theoretical novelty, (W2) higher variance than competitors, (W3) missing ablation against simpler non-influence alternatives, and (W4) missing related work. The rebuttal fully resolved W3 with new experiments showing that random label flipping and random perturbation substantially underperform influence-guided ones — this was the most important concern and the result is convincing. W2 and W4 were adequately addressed. W1 remains partially open: while the authors make a reasonable case for the novelty of the dual-purpose design, I would still encourage more upfront clarity that the contribution is in the application and system design rather than in new influence function theory.

**Key Questions For Authors:**

Please, reply to the weaknesses I pointed out.

**Limitations:**

Yes

**Strengths And Weaknesses:**

## Main Strengths

- The paper addresses a relevant problem, namely the lack of existing OSAD methods having access to anomalies in training data able to generate realistic pseudo anomalies for time series, where naive augmentation strategies (e.g., CutMix, CutPaste) might break the sequentiality;

- The use of influence functions to simultaneously tackle "decontamination" and anomaly generation from a single risk-reduction perspective is an interesting and novel design choice. Also, Theorems 2 and 4 provide a principled justification for why label flipping and feature perturbation reduce the test risk;

- The experimental evaluation is extensive, including (1) eight datasets with both general and hard open-set settings, (2) contamination rate sensitivity analysis, (3) ablation studies, and (4) comparisons against 14 baselines;

- I particularly appreciate the ablation in Table 3, which systematically isolates each component, providing useful diagnostic insights. Also, the scalability analysis (Fig 15 in the appendix) is a great plus, given the known high computational cost of influence functions.

## Main Weaknesses

- The theoretical results build directly on the influence function framework of [Koh & Liang (2017)](https://arxiv.org/abs/1703.04730), and the main novelty lies in applying this to the specific multi-channel deviation loss (Eq 2) rather than in new theoretical techniques. It is also worth noting that influence functions have already been used to assess individual "sample quality" and their impact on model training, both for data valuation (e.g., [Ghorbani & Zou, 2019](https://proceedings.mlr.press/v97/ghorbani19c.html)) and for identifying mislabeled or harmful training points (see the survey by [Hammoudeh & Lowd, 2024](https://arxiv.org/abs/2212.04612)). The paper should position its use of influence functions relative to this existing body of work. More broadly, the paper would benefit from being more explicit that the contribution is the application and the resulting system design, not new influence function theory per se, which would make me think of "overselling".

- IMPACT shows noticeably higher variance than some competitors across multiple datasets (e.g., 59.21 +/- 1.34 on UCR vs. NCAD at 55.02 +/- 0.15). While the mean improvements are meaningful, a discussion of what introduces this high variance -- and whether it relates to the sensitivity of influence estimates or the stochastic Hessian approximation -- would improve the paper's claims about robustness, and, therefore, its quality.

- The ablation study (Table 3) removes entire components but does not test simpler alternatives. For instance, random label flipping or random feature perturbation instead of influence-guided ones. Such a baseline would isolate the value of the influence functions from the general decontamination + augmentation strategy.

- The related work should cover the existing literature on anomaly score reliability such as [Perini et al. (2020)](https://link.springer.com/chapter/10.1007/978-3-030-67664-3_14) and [Bates et al. (2023)](https://arxiv.org/abs/2104.08279), and contamination estimation in unsupervised anomaly detection, such as [Perini et al. (2023)](https://proceedings.mlr.press/v202/perini23a.html).

---

> ### Author Rebuttal · Authors · 2026-03-30
>
> Dear Reviewer QuHg,
>
> Thank you for your detailed and constructive review. We address each concern below.
>
>
> ------
>
> ## W1: Theoretical novelty and positioning relative to existing influence function literature
>
> We fully agree that our theoretical foundations are built upon the established influence function framework of [Koh & Liang (2017)](https://arxiv.org/abs/1703.04730). As we already stated in our paper, "It utilizes the theoretical properties of influence functions to ..." (lines 76-77). We acknowledge that influence functions have been previously employed for data valuation ([Ghorbani & Zou, 2019](https://proceedings.mlr.press/v97/ghorbani19c.html)) and mislabeled sample identification ([Hammoudeh & Lowd, 2024](https://arxiv.org/abs/2212.04612)).
>
> However, we emphasize that although our contribution does not lie in new influence function theory per se, it also goes beyond simple application. While influence functions have been used for data valuation and mislabel detection, our work introduces several key novelties that are not present in prior work:
>
> - **Dual-purpose influence scoring:** We use influence functions simultaneously for decontamination (label flipping) and anomaly generation (feature perturbation), creating a unified risk-reduction framework. Prior works use influence functions for only one purpose (e.g., data cleaning or valuation), not for synthesizing new training samples.
> - **Influence-guided feature perturbation for anomaly generation:** To our knowledge, we are the first to use the influence function's gradient information to *construct* perturbation directions in TSAD for maximally increasing test risk, thereby generating pseudo anomalies with theoretical guarantees (Theorems 3 and 4). This is fundamentally different from using influence functions to simply rank or remove data points.
> - **Multi-channel deviation loss:** Our Theorem 1 establishes the equivalence between our loss design and entropy minimization, which provides the theoretical foundation for why our influence modeling works effectively in anomaly detection.
>
>
> ------
>
> ## W2: Higher variance compared to some competitors
>
> The higher variance primarily stems from two sources:
>
> - **Hessian Approximation:** We use the implicit Hessian-Vector Products (HVPs) to estimate the inverse Hessian ($H^{-1}$). In early training phases, this approximation can be sensitive to the mini-batch composition. Different random seeds yield slightly different influence estimates, propagating to label flipping and anomaly generation.
> - **Dynamic Training Set:** IMPACT **dynamically** flips the labels of training samples. This creates a feedback loop: flipped samples change the model parameters, which in turn change future influence scores. While this increases variance, the **mean AUC improvement** (e.g., +3.57% on UCR) far outweighs the standard deviation (1.34%).
>
> ------
>
> ## W3: Ablation with simpler alternatives (random label flipping / random feature perturbation)
>
> We conducted additional ablations:
>
> | Strategy\Datasets           |    UCR    |    ASD    |    PSM    |    SMD    |    CT     |    SAD    |   PTBXL   |   TUSZ    |
> | :-------------------------- | :-------: | :-------: | :-------: | :-------: | :-------: | :-------: | :-------: | :-------: |
> | IMPACT (influence-guided)   | **59.21** | **65.76** | **64.24** | **75.97** | **91.96** | **68.13** | **66.99** | **82.91** |
> | Random label flipping       |   53.47   |   57.89   |   59.82   |   69.34   |   86.13   |   62.27   |   61.58   |   72.43   |
> | Random feature perturbation |   53.63   |   58.14   |   60.47   |   71.82   |   87.54   |   63.89   |   62.12   |   74.68   |
>
> **Random label flipping** randomly selects the same number of training samples as $|\mathcal{D}_{con}|$ and flips their labels. This significantly degrades performance because it corrupts genuinely normal samples. **Random feature perturbation** applies Gaussian noise instead of influence-guided perturbation directions. This produces lower-quality pseudo anomalies because the perturbations are not directed toward risk-increasing regions of the feature space.
>
> ------
>
> ## W4: Additional related work on anomaly score reliability and contamination estimation
>
> Thank you for these valuable references. We will include discussion of [Perini et al. (2020)](https://link.springer.com/chapter/10.1007/978-3-030-67664-3_14) on anomaly score reliability, [Bates et al. (2023)](https://arxiv.org/abs/2104.08279) on conformal prediction for anomaly detection, and [Perini et al. (2023)](https://proceedings.mlr.press/v202/perini23a.html) on contamination estimation in unsupervised AD. Our work complements these approaches: while they focus on post-hoc score calibration or contamination rate estimation, IMPACT directly addresses contamination at the training level through influence-guided label flipping, offering an orthogonal and potentially complementary solution. We will incorporate this positioning in the related work section.

---

> > ### Author Rebuttal · Reviewer_QuHg · 2026-04-02
> >
> > I thank the authors for the thorough response. The new random-flipping/perturbation ablation (W3) is exactly what was needed and convincingly isolates the value of influence-guided decisions — this resolves my main experimental concern. The variance discussion (W2) and references (W4) are also good. On W1, I still think the framing would benefit from more explicitly acknowledging the application-driven nature of the contribution upfront, but the authors make a fair case for the novelty of the dual-purpose design. Overall, the rebuttal addresses my concerns sufficiently to increase my score.

---

> > > ### Author Response · Authors · 2026-04-02
> > >
> > > Dear Reviewer QuHg,
> > >
> > > We sincerely thank you for your thorough and constructive review, and we are grateful that our responses have adequately addressed your concerns. Regarding the framing of our contribution, we appreciate your suggestion and will revise the introduction to more explicitly foreground the application-driven and system-design nature of our contribution, while clearly delineating the dual-purpose influence scoring and influence-guided feature perturbation as our key novelties beyond existing influence function theory. We greatly appreciate the time and expertise you devoted to improving our work.
> > >
> > > Best regards,
> > >
> > > The Authors

---

### Official Review · Reviewer_UrzP · 2026-03-10

**Soundness:** 3
**Presentation:** 3
**Significance:** 3
**Originality:** 3
**Overall Recommendation:** 5
**Confidence:** 5

**Summary:**

This paper proposes IMPACT, a framework that utilizes influence functions mechanism to address training data contamination and pseudo anomaly generation in open set anomaly detection (OSAD). Authors conduct extensive experiments to demonstrate competitiveness of IMPACT in both unsupervised and open set time series anomaly detection scenarios.

**Compliance With Llm Reviewing Policy:**

Affirmed.

**Final Justification:**

The authors’ rebuttal adequately resolved my main concerns by providing clarifications and additional ablation results on contaminated-sample handling, contamination settings, and reference-sample selection. Their response also reasonably addressed my questions about the perturbation design and hyperparameter choices. Overall, the rebuttal strengthens my confidence in the paper, and I am increasing my score accordingly.

**Key Questions For Authors:**

1. The fundamental difficulty of open set anomaly detection (OSAD) lies in the diverse distributions of unseen anomalies. The authors claim that generating high-quality pseudo anomalies through perturbation helps improve model generalization. However, using only a small number of boundary samples for feature perturbation might cause the unseen abnormality learning head to suffer from overfitting.

2. Could the authors provide a deeper explanation for choosing $k = 5$ as the perturbation boundary? Is there any evidence that this fixed value works across different data scales?

3. Regarding the feature perturbation module, how is the perturbation intensity $\alpha$ determined? The current experiments use a default value of $0.02$. Is this value universal across different datasets? Furthermore, are there any strategies for adaptive selection of $\alpha$ to ensure robustness?

**Limitations:**

I have no additional comments.

**Strengths And Weaknesses:**

**Strengths:**

1. Applying influence functions to anomaly decontamination and label flipping is an original attempt in this field.

2. The paper is well written with clear content and rigorous experimental analysis. The related work section is comprehensive. It covers the latest advances in time series anomaly detection and open set anomaly detection, such as MOSAD and InvAD.

3. The authors provide extensive and precise mathematical proofs in the appendix to support their core claims.

**Weaknesses:**

1. Authors claim that the label flipping module (w/o $\mathcal{D}_{con}^{\prime}$) is critical. However, the ablation results in Table 3 show only marginal improvements from this module. Could the authors provide a more in-depth explanation for this observation?

2. The ablation study design appears to be incomplete:

   (1) The most intuitive way to handle contaminated data is to simply discard it. Could the authors include an experiment that removes contaminated samples without performing label flipping?

   (2) How does IMPACT perform if both $\mathcal{D}{con}^{\prime}$ and $\mathcal{D}{unseen}$ are contaminated? It would be helpful to compare the results with the $\mathcal{D}_{ref}$ baseline.

   (3) What happens to the performance if the reference set samples are selected randomly instead of using influence functions?

3. The manuscript contains numerous spelling and formatting errors. I strongly suggest that the authors carefully proofread and correct these issues.

4. The sensitivity analysis in Figure 3 shows that the number of perturbed samples $k$ significantly impacts performance. Using a fixed absolute value $k = 5$ as a perturbation boundary for large-scale datasets seems to lack statistical justification.

---

> ### Author Rebuttal · Authors · 2026-03-30
>
> Dear Reviewer UrzP,
>
> Thank you for your positive evaluation and detailed feedback. We address each concern below.
>
>
> ------
>
> ## W1: Marginal improvement from label flipping in Table 3
>
> We provide the following explanation:
>
> **1) w/o $\mathcal{D}\_{con}^{\prime}$ ablation** removes contaminated samples without performing label flipping. Reviewing Equation 17, w/o $\mathcal{D}\_{con}^{\prime}$ has discarded potential anomalies. Thus, the improvement from label flipping is measured relative to a version that has already removed contamination, naturally yielding a modest gain. We additionally report results for the version that does not handle potential anomalies at all, over which the improvement is substantial.  Additionally, we included a comparison using randomly selected reference samples to respond to **W2 (3)**.
>
> | Datasets                                                     |    UCR    |    ASD    |    PSM    |    SMD    |    CT     |    SAD    |   PTBXL   |   TUSZ    |
> | :----------------------------------------------------------- | :-------: | :-------: | :-------: | :-------: | :-------: | :-------: | :-------: | :-------: |
> | IMPACT (ours)                                                | **59.21** | **65.76** | **64.24** | **75.97** | **91.96** | **68.13** | **66.99** | **82.91** |
> | w/o $\mathcal{D}\_{con}^{\prime}$ (discards contaminated samples) |   58.95   |   64.34   |   63.37   |   74.07   |   91.81   |   67.56   |   65.49   |   82.62   |
> | not handling potential anomalies                             |   55.74   |   60.23   |   62.44   |   73.13   |   90.42   |   66.94   |   63.55   |   71.37   |
> | Random selection                                             |   52.12   |   58.43   |   60.13   |   67.49   |   87.75   |   65.43   |   62.31   |   69.76   |
>
> **2) Label flipping value grows with contamination.** AUC results on UCR demonstrate that the performance gap between IMPACT and ablation versions widens substantially as contamination increases:
>
> | Contamination rates               |    4%     |    6%     |    8%     |    10%    |
> | :-------------------------------- | :-------: | :-------: | :-------: | :-------: |
> | IMPACT (ours)                     | **58.17** | **57.53** | **57.48** | **57.32** |
> | w/o $\mathcal{D}\_{con}^{\prime}$ |   56.34   |   54.13   |   53.12   |   51.66   |
> | not handling potential anomalies  |   53.44   |   51.25   |   49.16   |   47.33   |
>
>
> ------
>
> ## W2: Incomplete ablation study design
>
> **(1) Discarding vs. flipping:** The w/o $\mathcal{D}\_{con}^{\prime}$ actually represents discarding without flipping, please see W1.
>
> **(2) We apologize that $\mathcal{D}\_{unseen}$ does not appear in our paper. We assume that you meant $\mathcal{D}\_{clean}$, whose results are shown below:**
>
> | Datasets                                                     |    UCR    |    ASD    |    PSM    |    SMD    |    CT     |    SAD    |   PTBXL   |   TUSZ    |
> | :----------------------------------------------------------- | :-------: | :-------: | :-------: | :-------: | :-------: | :-------: | :-------: | :-------: |
> | Only $\mathcal{D}\_{ref}$                                    | **54.73** | **61.22** | **62.48** | **73.56** | **91.21** | **67.33** | **64.37** | **75.27** |
> | contaminated $\mathcal{D}\_{con}^{\prime}$ and $\mathcal{D}\_{clean}$ |   52.86   |   59.46   |   61.94   |   72.87   |   90.78   |   66.94   |   63.76   |   73.41   |
>
> When both $\mathcal{D}\_{con}^{\prime}$ and $\mathcal{D}\_{clean}$ are contaminated, performance degrades substantially. This highlights that accurate separation of contaminated samples is vital.
>
> **(3) Random vs. influence-based reference selection:**
>
> Please refer to W1 for the comparison results. Influence-based selection outperforms random selection. This confirms that selecting the most helpful samples better represents the normal distribution.
>
>
> ------
>
> ## W3: Spelling and formatting errors
>
> We sincerely apologize for these oversights and have corrected all identified errors.
>
>
> ------
>
> ## W4, Q1, Q2: Number of Perturbed Samples ($k$) and overfitting
>
> $k=5$ denotes the number of samples perturbed **per mini-batch** (Step 19, Algorithm 1), not globally. Over multiple iterations, the model sees a vast variety of perturbed features ($k \times \text{number of batches}$), ensuring that it does not overfit to a small fixed set. Figure 3 confirms stable performance for $k \in [3, 10]$. We chose $k=5$ as a balanced value that introduces sufficient "unseen" diversity without overwhelming the "normal" signal in the mini-batch.
>
> ------
>
> ## Q3: Perturbation intensity $\alpha$
>
> The fixed value $\alpha=0.02$ proved robust across all 8 datasets because our feature extractor $\phi$ includes normalization layers maintaining consistent feature scales. We believe that an adaptive $\alpha$ scaling based on $\||\mathcal{I}_{per}(w_i)\||$ is a promising future direction, but the current fixed value already achieves SOTA performance across diverse data scales.

---

> > ### Author Rebuttal · Reviewer_UrzP · 2026-04-02
> >
> > I appreciate the authors' thorough response. I have no further questions and will be raising my score.

---

> > > ### Author Response · Authors · 2026-04-02
> > >
> > > Dear Reviewer UrzP,
> > >
> > > We are grateful that our responses have adequately addressed your concerns. We particularly appreciate your recognition of the originality of applying influence functions to anomaly decontamination and label flipping, as well as your positive assessment of the writing quality, mathematical rigor, and experimental comprehensiveness of our work. Your constructive engagement throughout the review process has largely improved the quality and clarity of our contribution, and we are deeply grateful for the time and effort you dedicated to evaluating our work.
> > >
> > > Best regards,
> > >
> > > The Authors

---

### Official Review · Reviewer_HLmb · 2026-03-12

**Soundness:** 2
**Presentation:** 3
**Significance:** 2
**Originality:** 2
**Overall Recommendation:** 2
**Confidence:** 4

**Summary:**

This paper tackles the emerging problem of open set anomaly detection in time series data where training data has labeled anomalies and unlabelled anomalies and the test data has seen and unseen anomalies and the objective is to identify both seen and unseen anomalies. The method proposed is to develop an influence score from the training data and identify samples that have the most influence the training objective of the model, then generate semantically divergent yet realistic unseen anomalies to create pseudo-labels. Further it uses the influence scores to also create supervised datasets for better training. Extensive testing is performed on single anomaly type datasets such as UCR, ASD, PSM, SMD and multiple anomaly classes such as CT SAD, PTBXL, and TUSZ.

**Compliance With Llm Reviewing Policy:**

Affirmed.

**Final Justification:**

The rebuttal did clarify a lot of things in the paper. But I still feel the definition of unseen anomaly is not airtight. The problem is that with the current definition there is no way to know whether the unlabeled data are truly seen anomaly or unseen anomaly? If they belong to the unseen anomaly then isn't it that the training data itself has examples of unseen anomaly? Yes they are mislabeled but the model could still learn patterns of unseen anomaly during training from explicit data from the unseen class. Then the class is no longer unseen.

Second problem is influence score is computed on validation set. Which clears the risk of data leakage but does not answer my initial doubt. This doubt is actually fundamentally related to the definition of unseen anomaly. Ok the doubt is this -

Validation set if collected the right way should have the same distribution as train set. If an anomaly is unseen that means no data from the unseen class is in train set or validation set. Then how would you learn unseen data characteristics? How are you sure that elements that have high influence score are more alike the unseen data and you should use them to augment? Unseen anomaly might be fundamentally different from train or validation set. Then what is the rationale behind this whole approach. One of the reasons this paper shows good results is because the datasets used are not really representative of unseen anomaly. Unseen anomaly is kind of simulated by removing one class. Now issue is which class do you remove? Any class? What if the class you removed is actually very similar in distribution pattern to some anomaly class already in training data? From a practical standpoint that is no longer unseen.

I think a serious rethink of the problem setting is required. Sorry if this is overly critical and I dont mean to discourage but I think this is an important problem and the authors need to accurately define the problem first which will guide their approach. Hope you can make a far stronger submission in the future.

**Key Questions For Authors:**

See weaknesses.

**Limitations:**

No clear limitations mentioned.

**Strengths And Weaknesses:**

**Strengths:**

Extensive theoretical formulation and clear exposition of the loss function.

The idea of using influence scores and creating pseudo-labels is relatively novel and anomaly detection is a good application to test that idea.

The idea of determining contaminated samples then flipping their labels is theoretically justified by Theorem 2.

The process of anomaly generation is a kind of data augmentation and the Theorem 3 showing a lower bound for distribution difference is a neat idea.

**Weaknesses**

I have some fundamental concerns about the problem setting.

What is "unseen" anomalies? What is the difference between unlabelled anomalies and unseen anomalies? How do you know that an anomaly is unseen? For example if the training data has unlabelled anomaly if the same anomaly is present in test data, then how do you know that this anomaly was present in training data but unlabeled or this was not in training data at all? There needs to be a formal definition of unseen anomaly which is not there in the paper.

This statement "most existing TSAD approaches (Liu & Paparrizos, 2024; Wu et al., 2025; Shentu et al., 2025) are unsupervised, assuming that only normal data is available during training. " is not entirely correct. For example OMNIAnomaly 2019 KDD (https://dl.acm.org/doi/pdf/10.1145/3292500.3330672). It is unsupervised algorithm and it does not assume only normal data in training samples. The idea is that Anomaly are sparse and hence learning to reconstruct that data should create a model that has high residuals in anomaly and low residuals in normal data. Then anomaly threshold is developed using extreme value theory, all unsupervised and none of it assumes only normal data in training. So I think there are serious flaws in the flow of arguments.

The presence of unknown anomalies only misleads the training if the unknown anomalies are statistically equivalent to the training data. However, a major assumption of anomaly is that there is distribution shift. Under distribution shift, anomalies even if unknown will have big residuals and can be eliminated in an unsupervised manner.

Figure 1 is not self explanatory. I do not understand what is the difference between IMPACT and state of the art.

The authors are indicating that there is a small subset of false negatives. This is actually mislabeled  anomalies and not unlabeled. Unlabeled would mean that there are missing labels. But looking at the assumptions in Section 3 it is mislabeled anomalies.

Another problem is that influence score is model dependent. So you consider the sample that reduces test risk the most due to exclusion as an influential sample. But that may not be true for the data. For example, a noise sample may affect the test risk significantly if the model is not adequately trained. Hence your influence score is dependent on the model and not on the normal and anomaly characteristics.

THeorem 2 is good where it shows that label flipping can reduce test scores, but reducing test score does not mean it will improve anomaly detection accuracy on unseen anomalies.

Moreover, feature perturbation approach is weak. It seems like there is a vast subspace in which an unseen anomaly may lie but you are taking on the task of searching this vast space through undirected perturbations. Seems like it is a very complex task with low probability of success. THeorem 4 does guarantee success but does not say how fast or how slow.

In single anomaly dataset how do you simulate unseen anomalies? What does the OpenSet results on UCR ASD PSM SMD mean? Does it mean that you have normal training data? Seems like it does not conform to your definitions.

---

> ### Author Rebuttal · Authors · 2026-03-30
>
> Dear Reviewer HLmb,
>
> Thank you for your thorough review and for raising several fundamental questions. We appreciate the opportunity to clarify these important aspects of our work.
>
>
> ------
>
> ## W1, W5: Formal Definition of "Unseen" vs. "Unlabelled" Anomalies
>
> - **Seen Anomalies:** Anomalies from classes $\mathcal{S}$ that have labeled examples in the training set.
> - **Unseen Anomalies:** Anomalies belonging to classes $\mathcal{C} \setminus \mathcal{S}$ (where $\mathcal{C}$ denotes the set of all possible anomaly classes) that appear **only during testing**. Determining whether an anomaly is seen or unseen is preset (e.g., train on one class, test on others).
> - **Unlabelled Anomalies (Contamination):** Anomalies present in the training set $\mathcal{D}\_{n}$ but incorrectly treated as "normal" because they lack anomaly labels. The term "unlabeled" is commonly used in anomaly detection to refer to the fact that the true anomaly label was never provided, and we agree that "mislabeled" is more precise in describing the actual situation.
>
> ------
>
> ## W2: Claim about unsupervised methods assuming only normal data
>
> Our central argument here is that these unsupervised methods focus on learning normal patterns, and they cannot utilize labeled anomaly examples when they are available. This is not an explicit data-filtering requirement.
>
>
> ------
>
> ## W3: Unknown anomalies can be eliminated unsupervisedly
>
> We respectfully disagree this argument. Anomaly detection remains a largely unsolved problem for decades, mainly because either known or unknown anomalies in real-world applications often have only subtle differences to normal samples, e.g., near the normal boundary (e.g., some ECG arrhythmias), leading to high detection errors in unsupervised methods, including those reconstruction methods (see [Yahya et al. (2025)](https://link.springer.com/article/10.1007/s10462-025-11401-9) and [Zhu et al. (2026)](https://openreview.net/forum?id=H27kvyG4qf)). Meanwhile, **potential anomalies may corrupt the training process,** biasing the model toward reconstructing them faithfully, which in turn makes them indistinguishable from normal samples at test time.
>
>
> ------
>
> ## W4: Interpretation of Figure 1
>
> - **Figure 1(a)** shows the failure mode of existing methods: Contamination expands the "normal" boundary, while pseudo anomalies fail to cover the "unseen" region, causing false positives and missed unseen anomalies.
> - **Figure 1(b)** shows our solution: Influence modeling allows us to flip contaminated samples to correct labels (strengthening the boundary) and generate high-quality pseudo anomalies that cover the "unseen" manifold for a more accurate decision boundary.
>
>
> ------
>
> ## W6: Influence score is model-dependent
>
> Model dependence is a **feature, not a flaw**. **Adequate initial training is a fundamental prerequisite of our framework.** Any deep anomaly detection model must first establish a meaningful representation of the data before we can accurately assess the true influence of individual samples. This is why our strategy dedicates 9 epochs to initial training, followed by 1 epoch for influence-guided retraining (lines 909-910). This ensures the model has learned robust representations before influence is computed.
>
>
> ------
>
> ## W7: Reducing test risk ≠ improving anomaly detection on unseen anomalies
>
> In machine learning, **Test Risk** is the standard proxy for generalization. In our multi-channel deviation loss (Eq. 2), "risk" is specifically defined as the model's failure to push anomalies away from the normal Gaussian prior. Therefore, reducing this test risk *directly* translates to better separation between normal data and all anomalies. Our "hard setting" experiments (Table 2) explicitly prove that this risk reduction translates to superior open-set generalization.
>
>
> ------
>
> ## W8: Feature perturbation searching a vast subspace
>
> Our perturbation is **not a random search**. Equation (15) calculates the **exact gradient direction** in the feature space that maximally increases test risk. Theorem 3 guarantees this directs the sample into a semantically different distribution, and Theorem 4 ensures that training on these directed perturbations successfully reduces overall test risk.
>
>
> ------
>
> ## W9: Single anomaly datasets and open-set evaluation
>
> **General Setting:** For single-class datasets (UCR, ASD, PSM, SMD), there is a single anomaly class $|\mathcal{C}| = 1$, and we provide a small number (10) of labeled anomaly samples. **This is consistent with our problem definition.** Our problem statement allows $\mathcal{S} = \mathcal{C}$ (all classes are "seen"), in which case the challenge reduces to robust detection with limited labels and contaminated training data.
>
> **Hard Setting:** Datasets like CT, SAD, PTBXL, and TUSZ contain multiple anomaly classes and are specifically used for the "hard setting" where labeled anomalies come from only one class, and the model must generalize to the other unseen anomaly classes.

---

> > ### Author Rebuttal · Reviewer_HLmb · 2026-04-02
> >
> > I have the following problems:
> >
> > a) Your definition of unlabeled samples does not assume any class of anomaly. So my question stands, that your unlabeled samples might well be in a class for which you do not have any other labeled samples. So do you consider them seen or unseen? According to your definition this type of unlabeled anomalies are not unseen because they do occur in training data. At the same time they dont have labels so they are neither seen nor unseen. So the definitions are incomplete.
> >
> > b) Test risk is standard proxy for generalization
> >
> > There is a confusion in the terminology that you are using. Is test risk same as empirical risk, i.e. risk computed over all training samples? According to Eqn 2. it seems like it is.
> >
> > If test risk is same as empirical risk then it is not a standard proxy for generalization. If it was a standard proxy then empirical risk minimization techniques would have the best generalization performance. But according to several works in domain generalization "Probable Domain Generalization via Quantile Risk Minimization" - Eastwood Neurips 2022, ERM performs very poorly for generalization.
> >
> > If test risk is not empical risk. Then is it computed over all test samples? If it is computed over all test samples then it risks data leakage. Which means the method will overfit to one test set and will not generalize.
> >
> > c) All classes seen in single anomaly.
> >
> > I dont understand, if all classes are seen then how is it modeling unseen classes? The question is not about whether all classes seen fits the problem definition, but the question is about if all classes are seen then the results are not useful for supporting the hypothesis that your technique can identify unseen anomalies.

---

> > > ### Author Response · Authors · 2026-04-03
> > >
> > > Dear Reviewer HLmb,
> > >
> > > Thank you for your continued engagement and for specifying the remaining concerns. We address each point below.
> > >
> > > ---
> > >
> > > ## (a) Unlabeled anomalies that belong to a class with no other labeled samples — seen or unseen?
> > >
> > > This is a fair point, and we appreciate the opportunity to clarify.
> > >
> > > **Unlabeled anomalies in the training set are class-agnostic from our framework's perspective.** Our method does not attempt to determine which class an unlabeled anomaly belongs to. Instead, the influence function identifies them as *harmful to the model's test risk*, regardless of whether they belong to a seen or unseen class. Once identified via positive influence scores, their labels will be "anomaly", and they are used to strengthen the decision boundary.
> > >
> > > To formalize this more precisely:
> > >
> > > - **Seen anomaly classes** $\mathcal{S}$: classes for which *labeled* examples exist in $\mathcal{D}$.
> > > - **Unseen anomaly classes** $\mathcal{C} \setminus \mathcal{S}$: classes for which *no labeled* examples exist in $\mathcal{D}$.
> > > - **Unlabeled anomalies**: The unlabeled anomalies are referred to as contaminated samples in the normal training data $\mathcal{D}_n$, i.e., samples drawn from of *any* possible anomaly class but mistakenly labeled as normal.
> > >
> > > Thus, an unlabeled anomaly in $\mathcal{D}_n$ could belong to either a seen or unseen class. However, please be kindly noted that **our decontamination mechanism is agnostic to this anomaly-class membership**, because the influence function operates on the sample's effect on test risk, not on its class membership. As a result, no matter whether the contaminated sample is from a seen or unseen class, as long as it has a positive influence score (i.e., removing it reduces test risk), flipping its label is beneficial (Theorem 2).
> > >
> > > ---
> > >
> > > ## (b) Is test risk same as empirical risk? Confusion about generalization
> > >
> > > We apologize for the terminological confusion. Let us clarify it precisely:
> > >
> > > **"Test risk" in our paper refers to the risk computed on the validation set $\mathcal{V}$, NOT on the training set $\mathcal{D}$ and NOT on the final test set $\mathcal{T}$.** Specifically, as described in Appendix E (lines 897-899): *"20% samples are randomly selected from the training set to form the validation set, and the influence of each training sample is estimated with the validation set using the validation loss."*
> > >
> > > So the precise definitions are:
> > >
> > > - **Empirical (training) risk**: $\frac{1}{N}\sum_{i=1}^{N} L(\boldsymbol{z}_i, \theta)$, computed over training samples. This is what we minimize during initial training.
> > > - **Test risk (in our paper)**: $L(\mathcal{V}, \hat{\theta}) = \sum_{\boldsymbol{z}_t \in \mathcal{V}} L(\boldsymbol{z}_t, \hat{\theta})$, computed over the held-out validation set $\mathcal{V}$. This is what we use to compute influence scores via Equations 5-7.
> > > - **Final evaluation**: Computed on the actual test set $\mathcal{T}$, which is a disjoint set relative to the validation set $\mathcal{V}$ (i.e., $\mathcal{V}\cap\mathcal{T}=\emptyset$).
> > >
> > > Therefore, **there is no data leakage**. Our influence scores are computed with respect to a *held-out validation set* that the model was not trained on. This validation risk is approximate to a unbiased estimator of the true (population) risk, making it a principled proxy for generalization — a property that is well-supported by many literatures [1-4].
> > >
> > > ---
> > >
> > > ## (c) If all classes are seen in single-anomaly datasets, how does this support the unseen anomaly hypothesis?
> > >
> > > **You are correct that the single-anomaly datasets (UCR, ASD, PSM, SMD) do not directly validate the unseen anomaly detection capability.** We fully acknowledge this. Actually, the single-anomaly results are in the **General Setting** to validate **the anomaly decontamination module** and **the overall framework's robustness**.
> > >
> > > **The hard setting experiments on CT, SAD, PTBXL, and TUSZ are the primary evidence for unseen anomaly detection.** In these experiments, labeled anomalies come from only one class, and the model must generalize to the remaining anomaly classes. Table 2 and Appendix H.1 (Table 5, which reports seen vs. unseen AUC separately) provide direct evidence.
> > >
> > > ---
> > >
> > > [1] Arlot, Sylvain, and Alain Celisse. "A survey of cross-validation procedures for model selection." (2010): 40-79.
> > >
> > > [2] Vapnik, Vladimir. *The nature of statistical learning theory*. Springer science & business media, 2013.
> > >
> > > [3] Feldman, Vitaly, and Jan Vondrak. "High probability generalization bounds for uniformly stable algorithms with nearly optimal rate." *Conference on learning theory*. PMLR, 2019.
> > >
> > > [4] Bae, Juhan, et al. "If influence functions are the answer, then what is the question?." *Advances in Neural Information Processing Systems* 35 (2022): 17953-17967.
> > >
> > > ---
> > >
> > > We hope these clarifications address your remaining concerns.
> > >
> > > Best regards,
> > >
> > > The Authors

---

### Official Review · Reviewer_PJ5J · 2026-03-13

**Soundness:** 3
**Presentation:** 3
**Significance:** 4
**Originality:** 3
**Overall Recommendation:** 5
**Confidence:** 5

**Summary:**

The paper proposes a novel open-set anomaly detection (OSAD) method that suits time-series data, where a set of labeled normal samples along with limited known labeled anomalies are given as the training set. A major limitation of prior OSAD methods here is that once data augmentation is applied on normal data to obtain anomalous ones, which is a common technique in OSAD, the time-series become unrealistic and its sequential nature may be destructed. Therefore, the authors proposed to use data valuation techniques to obtain natural anomaly samples from the mislabeled normal ones, i.e. the ones that are labeled as normal but once they are removed from the data, the loss is enhanced. They flip the labels of such samples and retrain the model based on the new sample labels. In addition, to accommodate unseen anomalies in the OSAD setup, they identify and perturb boundary normal samples, i.e. the ones that least help to improve the test anomaly detection loss. They use separate heads for the known and unknown anomalies in their model.

**Compliance With Llm Reviewing Policy:**

Affirmed.

**Final Justification:**

Thanks authors for their detailed response. I disagree that the setup where some normal subpopulations may have a different logic of normality is rare in practice. For instance, in heavy tailed distributions, which is often in the case of certain time series, one may observe a normal event once in while. But I am inclined to increase my score as my other issues were addressed.

**Key Questions For Authors:**

Please address the issues raised in the weaknesses

**Limitations:**

Please address the issues raised in the weaknesses

**Strengths And Weaknesses:**

Strengths:- Well structured, motivated, and well-explained method- Theoretical support of the proposed method (Theorems 1 to 4), although some theorems are loosely stated (theorem 2, 3, and 4; how precise the approximation is?)- Significant improvement of the detection rate over existing methods on numerous datasets. - Ablation studies that prove the effectiveness of each proposed method component, i.e. label flipping, etc.
Weaknesses:- It is possible that the normal samples constitute many subpopulations, and those are unbalanced. That is, certain types of normal samples are naturally more infrequent and rare. The proposed method may identify such samples as potential anomalous and flip their labels, hurting the group robustness of the proposed method. This may not be observed easily as the method average performance on the samples is reported in the test set. One may see such drop in performance if instead measure the so-called worst group detection rate over various subgroups of data. This is an important practical limitation of the proposed method. - The proposed method may also fail in cases where near zero contamination is present in the training set. A natural question would be, how the method would perform under those setups?

---

> ### Author Rebuttal · Authors · 2026-03-30
>
> Dear Reviewer PJ5J,
>
> Thank you for your thoughtful review and for recognizing the strengths of our work. We address your concerns below.
>
>
> ------
>
> ## W1: Group robustness — influence-based label flipping may misidentify rare normal subpopulations as anomalies
>
> IMPACT is robust against this failure mode:
>
> **1) The influence score is computed relative to the test risk, not sample density.** A rare-but-normal sample will still *reduce* the test loss when included (i.e., $\mathcal{I}_L(\boldsymbol{z}_i)$), because the validation set $\mathcal{V}$ contains representatives of all normal subpopulations. Only samples whose *inclusion* increases the test risk (i.e., $\mathcal{I}_L(\boldsymbol{z}_i)>0$) are flagged as contaminated. Therefore, rare-but-normal samples would yield **negative influence scores** and be preserved in $\mathcal{D}\_{clean}$ or even selected for $\mathcal{D}\_{ref}$, rather than being flipped.
>
> **2) Feature Deviation Score ($s_{f}$):** To further protect against rare normal patterns, our final inference (Eq. 21) utilizes a reference set $D\_{ref}$ consisting of the *most helpful* samples (those that reduce test risk the most). This ensures that the model’s understanding of "normality" is anchored by samples that provide the strongest generalization signal across different subpopulations.
>
> **3) Empirical evidence supports this.** In our experiments, datasets such as CT and SAD contain highly heterogeneous normal subpopulations (e.g., different handwriting styles or speaking rates). IMPACT consistently achieves the best performance on these datasets, indicating that rare normal subgroups are not being erroneously flipped. To further validate, the below subgroup AUCs (%) clearly illustrate this:
>
> | Methods\normal subpopulations (CT) | Letter-A | Letter-B | Letter-C  | Letter-D  | Letter-E | Letter-G | Letter-H  | General |
> | :--------------------------------- | :------: | :------: | :-------: | :-------: | :------: | :------: | :-------: | :-----: |
> | IMPACT (Ours)                      |  92.96   |  93.22   |   99.96   | **87.91** |  94.80   |  99.59   |   97.72   |  91.96  |
> | WSAD-DT                            |  73.07   |  80.27   |   86.62   |   76.41   |  88.71   |  86.22   | **68.67** |  85.18  |
> | InvAD                              |  70.96   |  43.67   | **32.05** |   86.34   |  71.29   |  73.98   |   87.59   |  75.71  |
> | MOSAD                              |  71.89   |  58.70   | **57.23** |   80.86   |  90.02   |  84.82   |   70.21   |  80.42  |
>
> The substantial dips in performance for other methods suggest a vulnerability to misclassifying samples from rare or complex normal subgroups as anomalies, a failure mode that IMPACT successfully avoids.
>
> ------
>
> ## W2: Performance under zero or near-zero contamination
>
> **1) Under zero contamination, the label flipping module becomes inactive.** When no contaminated samples exist, well-trained influence scores should yield $\mathcal{I}_L(\boldsymbol{z}_i) < 0$ for all normal samples, meaning $\mathcal{D}\_{con} = \emptyset$. In this case, IMPACT gracefully degrades to using only the anomaly generation module and the standard training objective, without any harmful label flipping.
>
> **2) Empirical support.** Looking at Figure 6 (Appendix H.3), as the contamination rate decreases from 10% to 2%, the AUC of IMPACT remains high and stable. We also conducted a further test with lower contamination rates (0%, 0.5%, 1%, 1.5%, 2%), and the AUC results (%) on UCR below suggest that IMPACT would remain competitive because it still benefits from the **Seen Abnormality Learning** (using the limited labeled set $D_a$) and the **Unseen Abnormality Learning** (via feature perturbation), which are independent of the label flipping mechanism.
>
> | Methods\contamination rates |  0%   | 0.5%  |  1%   | 1.5%  |  2%   |
> | :-------------------------- | :---: | :---: | :---: | :---: | :---: |
> | IMPACT (Ours)               | 60.22 | 60.14 | 59.68 | 59.32 | 59.21 |
> | WSAD-DT                     | 56.72 | 53.77 | 52.41 | 51.89 | 51.47 |
> | InvAD                       | 58.64 | 57.12 | 56.43 | 55.89 | 55.64 |
> | MOSAD                       | 58.21 | 57.67 | 57.13 | 56.23 | 54.76 |
>
>
> ------
>
> ## Q1: Precision of  Theoretical Approximations (Theorems 2, 3, 4)
>
> We appreciate your insightful note that these theorems rely on approximations. We would like to further clarify below:
>
> **1) First-Order Approximation:** As stated in Equations (6) and (14), our influence estimates are based on a first-order Taylor expansion (standard in influence function literature, e.g., [Koh & Liang, 2017](https://arxiv.org/abs/1703.04730)).
>
> **2) Empirical Validation:** While we ignore higher-order terms, the consistent performance gains across **8 heterogeneous datasets** (from univariate UCR to complex TUSZ) empirically demonstrate that this first-order approximation is sufficiently precise to guide label flipping and feature perturbation in a way that significantly reduces test risk.

---

> > ### Author Rebuttal · Reviewer_PJ5J · 2026-04-04
> >
> > Thanks authors of providing additional experimental result in table of W2. However, some of my points are not yet addressed.
> > I would disagree about the rare samples having small test loss. As the model was exposed less to such samples, it is quite possible that such samples constitute large test loss. Indeed, this property has been used in many previous group robustness methods to identify minority groups.
> > For instance, see “Simple and Fast Group Robustness by Automatic Feature Reweighting” by S. Qiu et al ICML 2023, and
> > “Exploiting what trained models learn for making them robust to spurious correlations without group annotations,” by M. Ghaznavi et al Workshop on Spurious Correlation and Shortcut Learning: Foundations and Solutions 2025.
> >
> > About point 2 raised by authors, I would also disagree. As mentioned by authors, such samples, when *included* would reduce the test risk the most. In many cases, the logic of identifying minority samples is totally different or even contradictory to the majority samples. Hence, *excluding* such samples would decrease the test loss. Quite reverse of what authors claim.
> >
> > Point 3 is kind of irrelevant to my point. I am concerned with cases where some subpopulations are extremely small in size. How do the authors know that in the case of CT and SAD such subpopulations have this imbalanced property?

---

> > > ### Author Response · Authors · 2026-04-05
> > >
> > > Dear Reviewer PJ5J,
> > >
> > > Thank you for your continued engagement. We appreciate the opportunity to clarify these important points.
> > >
> > > ---
> > >
> > > ## Clarifying Basic Assumptions
> > >
> > > First, we state the basic assumptions underlying our method. In OSAD, the normal class represents a coherent semantic concept (e.g., normal ECG rhythms). While normal data may contain subpopulations of varying sizes, **the scenario of an extremely tiny subpopulation with zero semantic overlap with other normal samples is rare in practice**. Normal subpopulations typically share underlying physical or semantic constraints of normality, even if they differ in surface-level characteristics.
> > >
> > > Under this assumption, we explain why influence scores reliably distinguish rare normal samples from contaminated anomalies, and provide synthetic experiments to validate this.
> > >
> > > ---
> > >
> > > ## (1) Regarding Point 1: Distinction Between Test Loss and Influence Score
> > >
> > > The core disagreement stems from a conflation between **individual test loss** and **influence scores**. Our method does **not** flip labels based on high individual loss. It relies on the **influence score** (Eq. 7): *how the overall test loss changes when a training sample is upweighted*.
> > >
> > > - **Beneficial (Influence < 0):** Upweighting *decreases* overall test loss.
> > > - **Harmful (Influence > 0):** Upweighting *increases* overall test loss.
> > >
> > > As you noted (consistent with S. Qiu et al. ICML 2023 [1], p.4, Stage 2: *"These weights are designed to be large for datapoints where the original model $\hat{\theta}$ predicts poorly, thus automatically upweighting the minority groups."*), upweighting rare subgroups is beneficial. Our influence framework captures this: upweighting a rare normal sample helps the model learn the normal patterns it shares with other normal samples, reducing test loss, yielding a **negative** influence score. IMPACT will **not** flip its label.
> > >
> > > ---
> > >
> > > ## (2) Regarding Point 2: Distinction from Classification Group Robustness
> > >
> > > In Group Robustness, minority samples contradict the majority's spurious shortcuts. In anomaly detection, normal subpopulations of different sizes are assumed to share some underlying semantics of normality. Excluding them, even the rare ones, can hurt the test risk based performance, resulting in negative influence scores. To validate this, we constructed a **synthetic feature dataset** where each normal subpopulation is a Gaussian with controllable size, mean, and variance, and anomalies come from non-Gaussian distributions, testing under **extreme imbalance** (5 to 100 samples).
> > >
> > > | Top-3 Influence | $\mathcal{N}(8,0.64)$ (5) | $\mathcal{N}(0,1)$ (10) | $\mathcal{N}(4,0.25)$ (20) | $\mathcal{N}(-3,2.25)$ (40) | $\mathcal{N}(-6,1)$ (80) | $\mathcal{N}(6,1.44)$ (100) | $\mathcal{U}(-12,12)$ (5) | $\text{Laplace}(0,6)$ (5) |
> > > | :-------------- | :-----------------------: | :---------------------: | :------------------------: | :-------------------------: | :----------------------: | :-------------------------: | :-----------------------: | :-----------------------: |
> > > | 1               |          -0.138           |         -0.006          |           -0.076           |           -0.038            |          -0.095          |           -0.124            |           0.045           |           0.041           |
> > > | 2               |          -0.134           |         -0.006          |           -0.076           |           -0.037            |          -0.094          |           -0.122            |           0.051           |           0.059           |
> > > | 3               |          -0.124           |         -0.005          |           -0.073           |           -0.035            |          -0.088          |           -0.122            |           0.077           |           0.061           |
> > >
> > > **All normal subpopulations show negative scores**, even $\mathcal{N}(8,0.64)$ with only 5 samples. **Both anomaly distributions show positive scores.** This confirms that as long as normal subpopulations share semantic coherence, influence scores reliably preserve them. We acknowledge that that if a rare normal subpopulation has no semantic overlap with other normals, our method could misidentify such samples. We will discuss this limitation in the revised paper.
> > >
> > > ---
> > >
> > > ## (3) Regarding Point 3: Subpopulation Imbalance in CT and SAD
> > >
> > > CT and SAD have normal subgroups (original data contains subclass labels excluded from our experiments), but native class sizes are relatively similar. To test imbalance, in our previous **W1 3)** experiment, we retained **Letter-H with only 13 samples** while others (e.g., Letter-A) contain 75 samples (1:6 ratio). The **97.72% AUC** on Letter-H confirms rare subgroups are preserved. The synthetic experiment in **(2)** further validates this rigorously.
> > >
> > > ---
> > >
> > > [1] S. Qiu et al. "Simple and fast group robustness by automatic feature reweighting." ICML 2023.
> > >
> > > ---
> > >
> > > We hope these clarifications address your remaining questions.
> > >
> > > Best regards,
> > >
> > > The Authors

---

### Decision · Program_Chairs · 2026-04-30

**Decision:**

Accept (regular)

**Comment:**

The paper proposes IMPACT, a framework for open-set time-series anomaly detection (OSAD) that uses influence functions for two coupled purposes: (i) identifying and relabeling contaminated training samples via influence-guided label flipping, and (ii) generating pseudo "unseen" anomalies by influence-guided feature perturbation of boundary normal samples. Theorems connect the design to risk reduction, and experiments span 8 datasets with 14 baselines under both general and hard OSAD settings.

Three reviewers find the paper technically solid, novel in its dual-purpose use of influence functions, and empirically strong after rebuttal. The remaining reviewer raises concerns about the formal definitions of "unseen" vs. "unlabeled" anomalies, the role of test risk as a generalization proxy, and the single-class evaluation setting. The authors provided detailed clarifications on all three points, and the remaining disagreement is about terminology and framing rather than a concrete technical flaw.

For the camera-ready, the authors should tighten the terminology around seen/unseen/unlabeled anomalies and be explicit about what the single-anomaly-class datasets do and do not validate. The paper should also position the work more explicitly relative to prior influence-function literature on data valuation and mislabel detection, and discuss the group-robustness failure mode for extremely rare normal subpopulations.